# Adjunctive phage therapy improves antibiotic treatment of ventilator-associated-pneumonia with *Pseudomonas aeruginosa*

Chantal Weissfuss [1] ✉, Jingjing Li[1], Ulrike Behrendt[1], Karen Hoffmann[1], Magdalena Bürkle [1], Chunjiang Tan[1], Gopinath Krishnamoorthy[1], Imke H. E. Korf[2], Christine Rohde[3], Baptiste Gaborieau [4,5,6], Laurent Debarbieux [4], Jean-Damien Ricard [5,6], Martin Witzenrath[1,7], Matthias Felten[1,8] & Geraldine Nouailles [1,8] ✉

Bacterial multidrug resistance poses an urgent challenge for the treatment of critically ill patients developing ventilator-associated pneumonia (VAP). Phage therapy, a potential alternative when conventional antibiotics fail, has been unsuccessful in first clinical trials when used alone. Whether combining antibiotics with phages may enhance effectiveness remains to be tested in experimental models. Here, we use a murine model of *Pseudomonas*-induced VAP to compare the efficacy of adjunctive phage cocktail for antibiotic therapy to either meropenem or phages alone. Combined treatment in murine VAP results in faster clinical improvement and prevents lung epithelial cell damage. Using human primary epithelial cells to dissect these synergistic effects, we find that adjunctive phage therapy reduces the minimum effective concentration of meropenem and prevents resistance development against both treatments. These findings suggest adjunctive phage therapy represents a promising treatment for MDR-induced VAP, enhancing the effectiveness of both antibiotics and phages while reducing adverse effects.

Ventilator-associated pneumonia (VAP) is one of the most common hospital-acquired infections[1]. VAP is defined as pneumonia occurring in hospitalized patients that have been maintained on a mechanical ventilator for at least 48 h. As a result, VAP affects the most vulnerable and critically ill patients in intensive care units. It extends the time on mechanical ventilation and hospital stay, thereby increasing both morbidity and healthcare costs, and is associated with high mortality rates ranging from 13%[2] to 42%[3] among ICU patients or even 47%[4] among COVID-19 patients in recent years. Treatment is driven by the

patient's risk factors, like general comorbidities or illness severity[1]. Although carbapenems, e.g., meropenem, are recommended as a first-line treatment for VAP when antibiotic resistance is suspected[5], their use carries the risk of selecting for carbapenem-resistant pathogens, including *Pseudomonas (P.) aeruginosa*, *Acinetobacter baumanii*, *Klebsiella pneumoniae* and *Enterobacter* species[6,7]. To avoid carbapenem resistance, other antibiotic classes or combinations of different antibiotics can be used, as various clinical trials and retrospective studies have shown[8–10]. Nevertheless, the non-specific action of antibiotics also

[1]Department of Infectious Diseases, Respiratory Medicine and Critical Care, Charité—Universitätsmedizin Berlin, Corporate Member of Freie Universität Berlin and Humboldt-Universität zu Berlin, Berlin, Germany. [2]Pharmaceutical Biotechnology, Fraunhofer Institute for Toxicology and Experimental Medicine, Braunschweig, Germany. [3]Leibniz Institute DSMZ-German Collection of Microorganisms and Cell Cultures, Braunschweig, Germany. [4]Institut Pasteur, Department of Microbiology, Bacteriophage Bacteria Host Laboratory, Université Paris Cité, CNRS UMR6047, Paris, France. [5]Infection Antimicrobials Modelling Evolution, Université Paris-Cité, Inserm, UMR 1137, Paris, France. [6]APHP, Hôpital Louis Mourier, DMU ESPRIT, Service de Médecine Intensive Réanimation, Colombes, France. [7]German Center for Lung Research (DZL), Berlin, Germany. [8]These authors jointly supervised this work: Matthias Felten, Geraldine Nouailles. ✉e-mail: chantal.weissfuss@charite.de; geraldine.nouailles@charite.de

destroys part of the patients' microbiota and causes other serious side effects. In addition, treatment is often complicated by the multidrug-resistance (MDR) status of the causative pathogens[1]. MDR bacteria have become increasingly common, especially in hospital settings[3]. A leading causative agent of VAP is carbapenem-resistant *P. aeruginosa*[1,11], a high-priority MDR pathogen, as categorized by the World Health Organization based on the urgency for new drug development[3,7]. Over the last decades, few truly novel antibiotic classes have been developed, most of which have been modifications to existing classes or combinations of antibiotics and non-antibiotic compounds[12,13]. One of the few counterexamples is a new class of antibiotic targeting the LpxH protein in gram-negative bacteria reported this year[14]. The need to develop new therapeutic approaches thus remains critical.

A potential alternative approach is therapy with bacteriophages (phages)[15,16], which are lytic viruses, that infect and lyse their bacterial hosts with high specificity[17]. Numerous experimental studies[18–22] and individual human case reports[23–27] have demonstrated the effectiveness of therapeutic phages for targeting MDR pathogens. Importantly, to date, no serious side effects have been reported with phage therapy[28–31] and phage therapy appears to not affect the patient's microbiota[32,33]. However, although case reports highlight the effectiveness in individual patients, no controlled clinical trial has yet demonstrated successful therapeutic application. A phase 1/2 trial[34], treating burn wounds infected by *P. aeruginosa* with a specific phage cocktail, was stopped due to insufficient efficacy of bacterial reduction likely because of the low phage titer. Intravesical phage therapy of urinary tract infections with a commercial phage cocktail[35] failed as microbiological treatment success rates did not differ between phage, antibiotic and placebo groups. In addition, a randomized trial with two different phage cocktails for treating acute bacterial diarrhea in children[36] did not show intestinal phage amplification and improvement in diarrhea outcome over standard therapy. However, none of these trials identified adverse events attributable to phage application. They did, however, highlight that the need for sufficiently high phage titers, phagograms for assessing target bacteria's susceptibility, and monitoring the emergence of phage-resistant clones will be crucial for success in future clinical trials.

A further caveat remains: as is the case for antibiotics, bacteria are likewise able to develop resistance against specific phages[37,38]. To overcome the limitations of phage and antibiotic therapy when used individually, treatment options using a combination of both are under discussion and have been explored in experimental disease models[39–41]. To gain more mechanistic insights and develop a phage-antibiotic combination strategy effective against MDR-induced VAP in the clinic, suitable preclinical animal models reflecting the clinical situation of the human disease are required. Only a few animal models for studying VAP have been reported, most of them using pigs[42,43]. Here, we use a novel preclinical murine VAP model as well as human in vitro cell models to demonstrate the effectiveness and (putative) synergy mechanisms of using phages as adjunctive therapy in combination with the antibiotic meropenem over classical monotherapy. In our previous study, we were able to exclude a significant immunogenicity of the phage cocktail in use and the effectiveness of the intraperitoneal administration route for treating MDR bacteria[31]. Phages used in our cocktail were previously selected for their effectiveness against ~55% of clinical *P. aeruginosa* isolates, including the *P. aeruginosa* strain PAO1, which was used in this study. We observed faster clinical improvement of mice with VAP receiving adjunctive phage therapy in vivo, which could be confirmed in vitro. Meropenem was effective at lower concentrations when combined with the phage cocktail and *Pseudomonas*-induced epithelial cell damage was prevented. These data reveal adjunctive phage treatment with antibiotic therapy as a promising and safe approach for treating MDR-induced VAP infections, which can enhance the effectiveness of antibiotics as well as phages, while reducing adverse effects.

## Results

### Adjunctive treatment of antibiotics with phages leads to faster clinical improvement of *P. aeruginosa*-induced VAP in mice than single treatments

To compare the effectiveness of adjunctive phage therapy to standard monotherapy, we used the antibiotic meropenem and a phage cocktail of two *Pseudomonas*-phages, JG005[44] and JG024[45], alone or in combination, to treat *Pseudomonas*-induced VAP in a murine model. C57BL/6J mice were ventilated with high tidal volume for 4 h[46,47] and subsequently infected intratracheally with *P. aeruginosa* strain PAO1 followed by extubation. Treatment occurred at 4 and 16 hpi with either meropenem (monotherapy; standard-of-care), the phage cocktail (monotherapy), a combination of both (adjunctive phage therapy), or PBS as control. The clinical status, reflecting disease burden–comprising both pathogen load and inflammation–was monitored at 4, 16, and 24 hpi, and mice were sacrificed at 24 hpi for analysis (Fig. 1a).

Up to 4 hpi, all mice underwent the same experimental procedure. At this time point, all groups exhibited a high clinical disease score. The disease scores improved upon treatment, as evidenced by a significantly more substantial score reduction in less time with phage treatment, especially with adjunctive treatment, compared to the meropenem-only group: down to 4 score points at 16 hpi and 1 score point at 24 hpi (Fig. 1b). In line with this, a more marked recovery of body temperature was observed in the two groups receiving phages, with significant improvement at 16 hpi and a continued upward trend at 24 hpi (Fig. 1c). This indicates a faster clinical improvement from VAP compared to the control and meropenem only groups. Contrary to prompt recovery in body temperature after treatment, body weight loss progressed in all groups and did not respond to therapy in the analyzed time frame (Fig. 1d). Further, we found phage titers to be similar in all organs sampled from the phage-treated groups (Fig. 1e), indicating successful phage distribution following the i. p. administration route, confirming previous findings[31]. All three antimicrobial treatments significantly reduced the bacterial burden locally (Fig. 1f) and systemically (Fig. 1g) compared to the untreated control. As expected, groups that received meropenem treatment alone or in combination therapy reduced the bacterial load most strongly, e. g. in lungs and BAL by 2–3 log levels, respectively. Taken together, mice that received adjunctive phage therapy benefitted the most, with significantly improved clinical scores compared to either meropenem or phage cocktail therapy alone.

Next, we investigated the immune response to VAP under the different treatments. As expected, *Pseudomonas*-induced VAP led to an increase in local IL-6 and other pro-inflammatory cytokine levels, including IL-23, TNFα, and IL-1α in BALF at 24 hpi (Fig. 2a; Supplementary Fig. S2). Following phage therapy, either alone or combined with meropenem, cytokine levels were significantly lower compared to the untreated control group. In contrast, monotherapy with meropenem resulted in a trend toward reduced levels of IL-6 and IL-23 and significantly lowered levels of TNFα and IL-1α compared to the control group. However, these levels remained higher compared to those observed in the phage-only and combination treatment groups. No significant changes in innate immune cells, including leukocytes, neutrophils, and alveolar macrophages, were seen in BAL and lungs in any treatment group (Fig. 2b–d; Supplementary Fig. S3).

As we observed positive treatment results in terms of clinical improvement with reduced levels of pulmonary inflammatory markers upon phage monotherapy and adjunctive phage treatment, we next investigated lung barrier damage and systemic organ injury. The release of the pro-inflammatory mediator IL-6 in plasma was significantly reduced with all three treatments (Fig. 3a). However, the

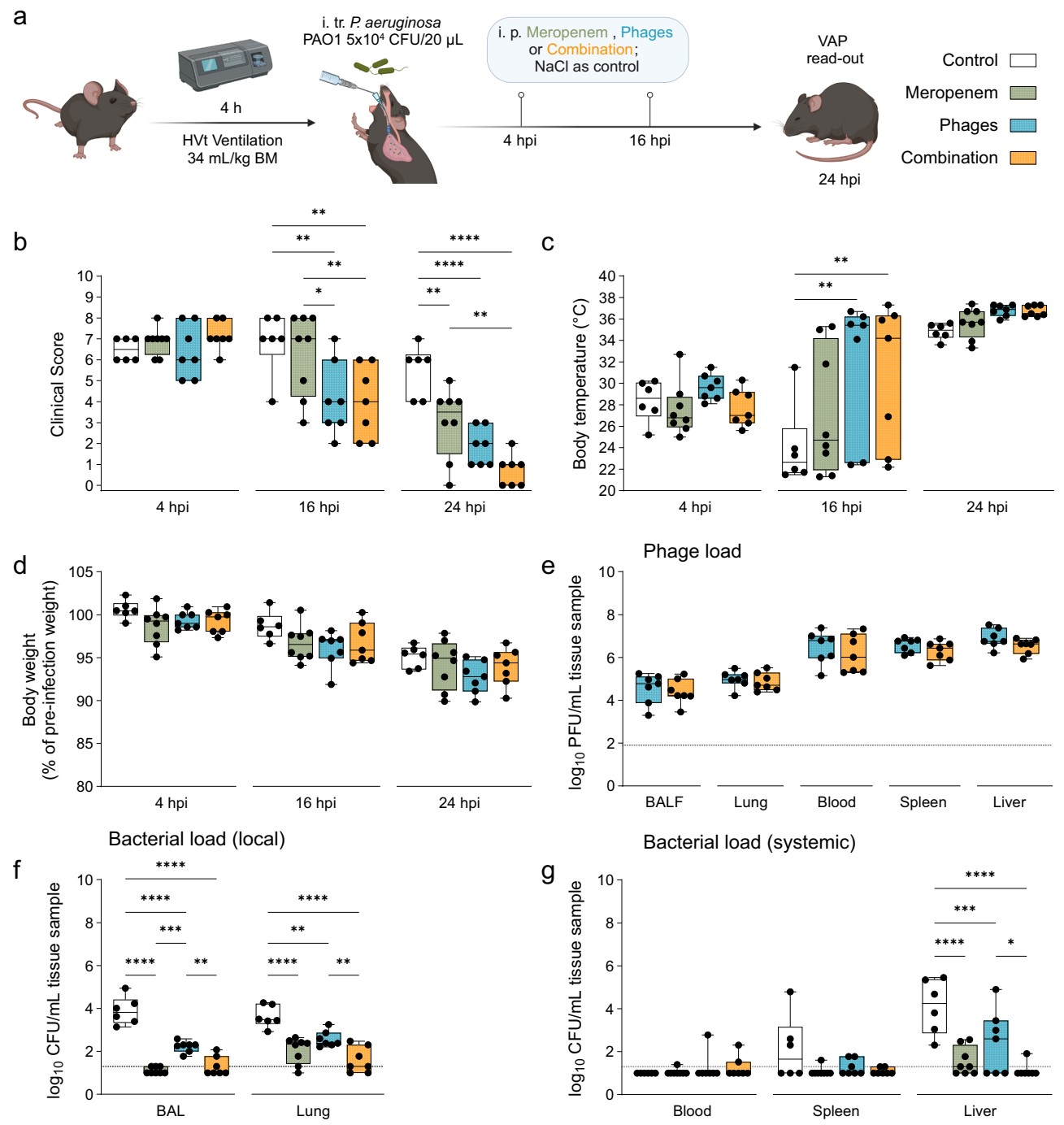

permeability index, as assessed by the murine serum albumin (MSA) ratio between BALF and plasma, was significantly reduced in the phage and combination groups, but not with meropenem alone (Fig. 3b). Thus, despite its observed antibacterial effect (see Fig. 1f, g) meropenem failed to prevent lung barrier injury. Next, we monitored if the phage therapy could also reduce multiorgan dysfunction observed during VAP. Liver and kidney injury were monitored though different clinical markers: plasma levels of alanine aminotransferase (ALT) were also significantly reduced with phage or combination therapy, while aspartate aminotransferase (AST), urea, triglyceride and the level of total bilirubin showed only a small trend or remained unchanged (Fig. 3c). The adjunctive phage therapy therefore offers no additional advantage over phage treatment alone in terms of indicators of organ damage but is more protective than meropenem therapy alone.

Taken together, these data demonstrate the potent efficacy of a combination therapy in our murine *Pseudomonas*-induced VAP model. Adjunctive phage treatment results in faster clinical improvement in murine VAP without a concomitant increase in pro-inflammatory cytokine release and prevents lung and organ injury, evidencing a benefit over monotherapy.

## Combination therapy is more effective against *P. aeruginosa* infection in vitro than antibacterial agents alone

To further resolve the underlying mechanisms that led to the improved outcome of VAP in mice treated with adjunctive phage therapy over antibiotic or phages alone, we conducted several in vitro studies to evaluate the impact of each treatment regarding their efficacy, kinetics and the cellular toxicity resulting from their different modes of bacterial lysis.

**Fig. 1 | Faster clinical improvement under phage treatment in a murine *P. aeruginosa*-induced VAP model. a** Experimental Plan created in BioRender. Nouailles, G. (2025) https://BioRender.com/c82e2a9. Naive mice (C57BL/6J WT, female, 8–10 weeks; Charles River) were ventilated for 4 h (h) under high tidal volume (HVt: 34 mL/kg body weight) triggering ventilator-induced lung injury. VAP was then induced by intratracheal (i. tr.) infection with PAO1 ($5 \times 10^4$ CFU/20 μL), followed by extubation and two intraperitoneal (i. p.) treatments at 4 h post infection (hpi) and 16 hpi with meropenem (10 mg/mouse per injection), a *Pseudomonas*-specific phage cocktail (~$5 \times 10^7$ PFU/phage per injection), or the combination of both. Control mice received NaCl (0.9 %) respectively. Analysis time point was 24 hpi. Graphs displaying (**b**) murine clinical disease score (see Supplementary Table S1), including **c** body temperature and **d** body weight change at indicated time points. Logarithmic display (Y = log(CFU + 1)) of **e** phage titers and **f, g** bacterial loads at 24 hpi in indicated organs (BAL, homogenized half lung (right), blood, spleen, liver). Results are shown as box plots depicting median, quartiles, and range, as determined by two-way ANOVA with Tukey's multiple comparisons test: *$p < 0.05$, **$p < 0.01$, ***$p < 0.001$, ****$p < 0.0001$; $n = 6$–8 mice per group (Control group $n = 6$; Meropenem group $n = 8$; Phages group $n = 7$;

Combination group $n = 7$). Color coding for groups: Control group: white, Meropenem group: green, Phages group: blue; Combination group: orange. Dotted line reflects detection limit. CFU colony-forming unit, HVt high tidal volume, hpi hours post infection, PFU plaque-forming unit, VAP ventilator-associated pneumonia. *P*-values were adjusted for multiple testing, exact *p*-values in order of appearance: **b** 16 hpi: Control vs Phages: $p = 0.029$, Control vs Combination: $p = 0.0016$, Meropenem vs Phages: $p = 0.0163$, Meropenem vs Combination: $p = 0.009$; 24 hpi: Control vs Meropenem: $p = 0.0048$, Control vs Phages: $p < 0.0001$, Control vs Combination: $p < 0.0001$, Meropenem vs Combination: $p = 0.0077$, **c** 16 hpi: Control vs Phages: $p = 0.0011$, Control vs Combination: $p = 0.0057$, **f** BAL: Control vs Meropenem: $p < 0.0001$, Control vs Phages: $p < 0.0001$, Control vs Combination: $p < 0.0001$, Meropenem vs Phages: $p = 0.0003$, Phages vs Combination: $p = 0.0043$, Lung: Control vs Meropenem: $p < 0.0001$, Control vs Phages: $p = 0.0012$, Control vs Combination: $p < 0.0001$, Phages vs Combination: $p = 0.002$, **g** Liver: Control vs Meropenem: $p < 0.000$, Control vs Phages: $p = 0.001$, Control vs Combination: $p < 0.000$, Phages vs Combination: $p = 0.0121$. Source data are provided as a Source Data file.

A common in vitro model of stretch-induced lung injury for studying the role of mechanical ventilation is the FlexCell system[47,48]. For an in vitro VAP model, we combined the cyclic stretch of human primary alveolar epithelial cells (HPAEpiC) with a subsequent infection with the *P. aeruginosa* strain PAO1 (MOI 1) 24 h later. The treatment occurred 2 hpi with meropenem (at either $5 \times$ MIC or $0.5 \times$ MIC), the *P. aeruginosa*-specific phage cocktail (at either MOI 0.5 or 0.005), or a combination of both (at doses ranging from high (++++: Meropenem $5 \times$ MIC, Phages MOI 0.5) to low (+: Meropenem $0.005 \times$ MIC, Phages MOI 0.005)). At 4 and 8 hpi, the treatment efficacy was evaluated (Fig. 4a).

First, we examined the bacterial load after the different treatments. Except at the lowest concentration tested, the adjunctive phage therapy demonstrated a greater antibacterial effect than either antibacterial agent used alone (Fig. 4b, Supplementary Tables S2–S4). Phage treatment worked faster in killing the bacteria than meropenem, resulting in a decreased bacterial load of up to 4 logs at 4 hpi (Phages MOI 0.5) compared to meropenem ($5 \times$ MIC), which only led to a decrease of up to 2 logs. However, phage monotherapy harbors the risk of outgrowth from phage-resistant bacterial clones over time. This could be seen in an increased bacterial load at 8 hpi compared to 4 hpi. Despite being less effective in reducing the bacterial burden, meropenem monotherapy maintained its antibacterial effect longer than phage treatment alone, evident by the further decrease in bacterial loads at 8 hpi. Notably, the adjunctive phage therapy combined the fast and specific killing effect observed with specific phages and the prolonged antibacterial potential of the standard-of-care treatment with meropenem, limiting the outgrowth of likely phage-resistant clones. Even with 10-fold lower concentrations than the lowest used single treatments (Combination ++: Meropenem $0.05 \times$ MIC, Phages MOI 0.05), the bacterial load at 8 hpi remained below the detection limit (Fig. 4b).

All treatments showed significantly reduced levels of IL-6 and IL-8 release by the epithelial cells at 8 hpi compared to infection control but there were no differences between any of the treatments (Fig. 4c). Neither did we observe any differences in the release of LDH (Fig. 4d), a common marker for cell death, between different treatments.

Next, we performed an in vitro infection assay to better understand the antibacterial kinetics of each therapeutic agent independently of putative defense responses by epithelial cells. For this purpose, liquid cultures of *P. aeruginosa* strain PAO1 (PAO1-LC) were incubated with different concentrations of meropenem, the phage cocktail, or a combination of both; at different time points (15, 30, 45, 60, 120, 180 min) samples were analyzed for bacterial loads (Fig. 5a).

The phage treatment reduced the bacterial load more quickly than meropenem and in a dose-dependent manner. Within the first 30 min of incubation, the highest concentration of phage cocktail (MOI 5) could significantly reduce the bacterial load to ~$10^2$ CFU/mL, whereas the highest concentration of meropenem ($50 \times$ MIC) could reduce bacterial load to ~$10^6$ CFU/mL, a minor reduction compared to the untreated control (~$10^8$ CFU/mL) (Fig. 5b, Supplementary Fig. S5b). However, as expected, a minor regrowth of likely phage-resistant bacterial clones under phage monotherapy appears after 60 min, regardless of the concentration. In this study, the efficacy of the phage cocktail was confirmed, regardless of the individual contributions of single phages, despite the likelihood of receptor competition or cross-resistance. The adjunctive phage therapy was most effective, including fast bacterial killing and prevention of regrowth of likely phage-resistant clones for up to 3 h in this experimental setting (Fig. 5b), matching results in the presence of epithelial cells (see Fig. 4b). In addition, a lower concentration of meropenem was effective when combined with the phage cocktail. After 60 min, the combination ++++ (Meropenem $5 \times$ MIC, Phages MOI 0.5) succeeded in reducing the bacterial load to below the detection limit, while a 10-fold higher dose of meropenem ($50 \times$ MIC) failed when given alone (Fig. 5b).

Even with an extended incubation time of 24 h, adjunctive phage treatment demonstrated the highest efficacy compared to monotherapies in vitro (Supplementary Fig. S5). In the high-dose combination treatment ++++ (Meropenem $5 \times$ MIC, Phages MOI 0.5) the bacterial load remained below the detection limit. In contrast, a slight increase in likely phage-resistant bacteria was observed with the mid-dose combination ++ (Meropenem $0.5 \times$ MIC, Phages MOI 0.05), but only after 9 h. However, the high-dose meropenem monotherapy ($50 \times$ MIC) failed to significantly reduce the bacterial load as effectively as the adjunctive therapy, even at later time points. At the 9 h-time point, bacterial concentration remained higher than under both combination therapies and only decreased to ~$10^3$ CFU/mL by 24 hpi. As expected, bacterial load still increased in a time-dependent manner under phage monotherapy (MOI 5). These findings highlight the superior efficacy of adjunctive phage treatment, enabling the use of reduced concentrations of both meropenem and the phage cocktail. As meropenem and phages may release different amounts of endotoxins due to their different antibacterial modes of action, we further quantified the released endotoxins in the supernatants of the infection assay for each treatment condition. Less free endotoxins were released upon phage treatment alone compared to meropenem alone, but no significant differences were seen after 60 min anymore (Fig. 5c). No additive effect of the adjunctive phage treatment was observed.

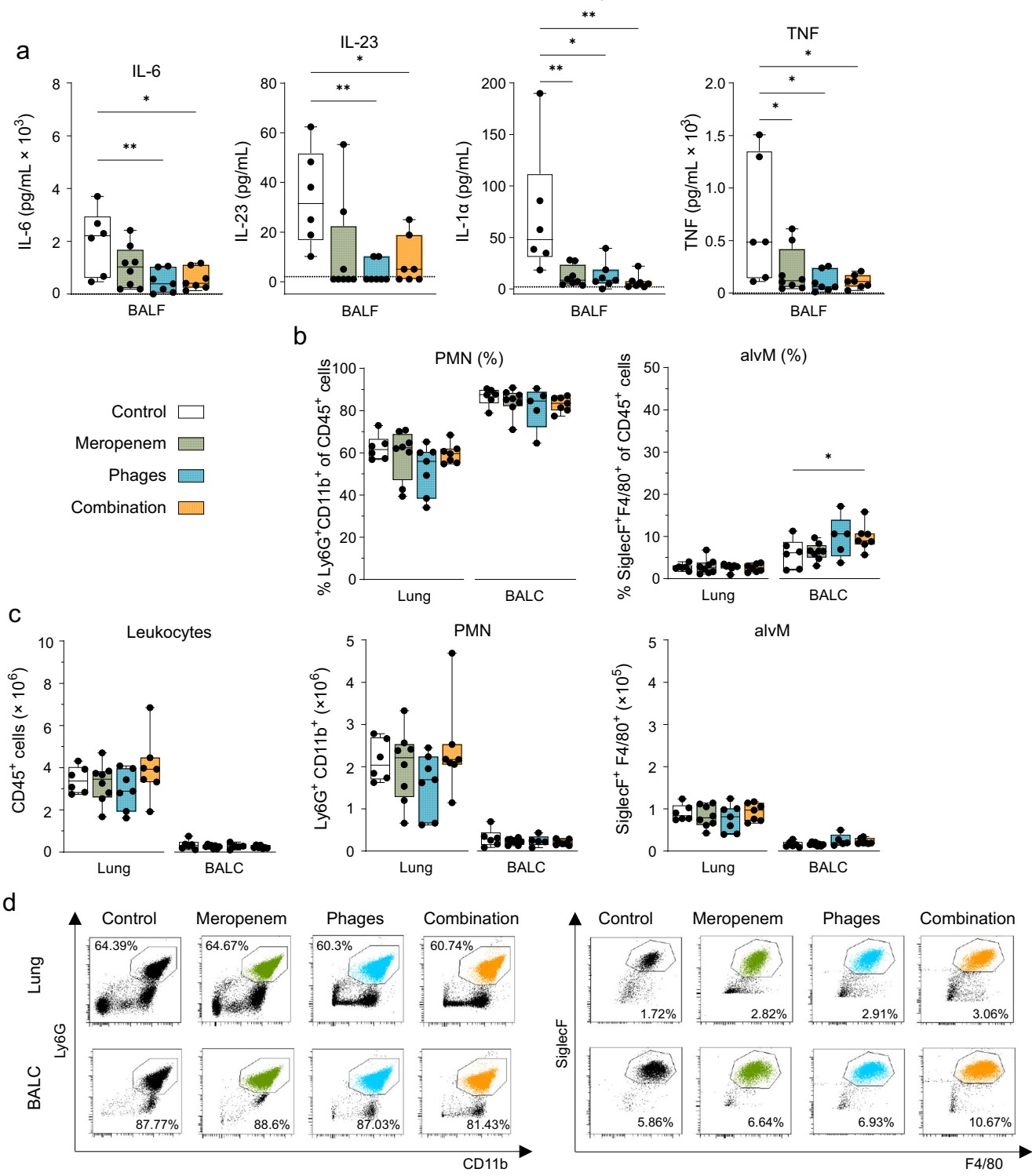

**Fig. 2 | Mice with *Pseudomonas*-induced VAP showed reduction in local inflammatory mediators, but not immune cells in alveolar spaces after phage treatment. a** Level of pro-inflammatory cytokines IL-6 (pg/mL), IL-23 (pg/mL), TNFα (pg/mL), and IL-1α (pg/mL) in BALF of mice with VAP analyzed at 24 hpi. **b** Percentages, **c** total cell numbers and **d** representative dot plots of leukocytes, polymorphonuclear leukocytes (PMN) and alveolar macrophages (alvM) in half lungs (left) and BALC of mice with VAP analyzed at 24 hpi by flow cytometry. For full gating strategy and cell type identifying markers see Supplementary Fig. S1. Results are shown as box plots depicting median, quartiles, and range, as determined by **a** ordinary one-way ANOVA or **b, c** two-way ANOVA with Tukey´s multiple comparisons test: *$p < 0.05$; **$p < 0.01$; $n = 6$–$8$ mice per group (Control group $n = 6$; Meropenem group $n = 8$; Phages group $n = 7$ (except Phages group BALC $n = 5$,

technical loss of $n = 2$); Combination group $n = 7$). Color coding for groups: Control group: white, Meropenem group: green, Phages group: blue; Combination group: orange. Dotted lines represent the detection limit, respectively. BALF, bronchoalveolar lavage fluid; BALC, BAL cells; IL, interleukin, VAP, ventilator-associated pneumonia. *P*-values were adjusted for multiple testing, exact *p*-values in order of appearance: **a** IL-6: Control vs. Phages: $p = 0.0082$, Control vs. Combination: $p = 0.0143$, IL-23: Control vs. Phages: $p = 0.007$, Control vs. Combination: $p = 0.0247$, IL-1α: Control vs. Meropenem: $p = 0.0073$, Control vs. Phages: $p = 0.01$, Control vs. Combination: $p = 0.0043$, TNF: Control vs. Meropenem: $p = 0.0412$, Control vs. Phages: $p = 0.0123$, Control vs. Combination: $p = 0.0126$. **b** alvM (%): Control vs. Combination: $p = 0.0364$. Source data are provided as a Source Data file.

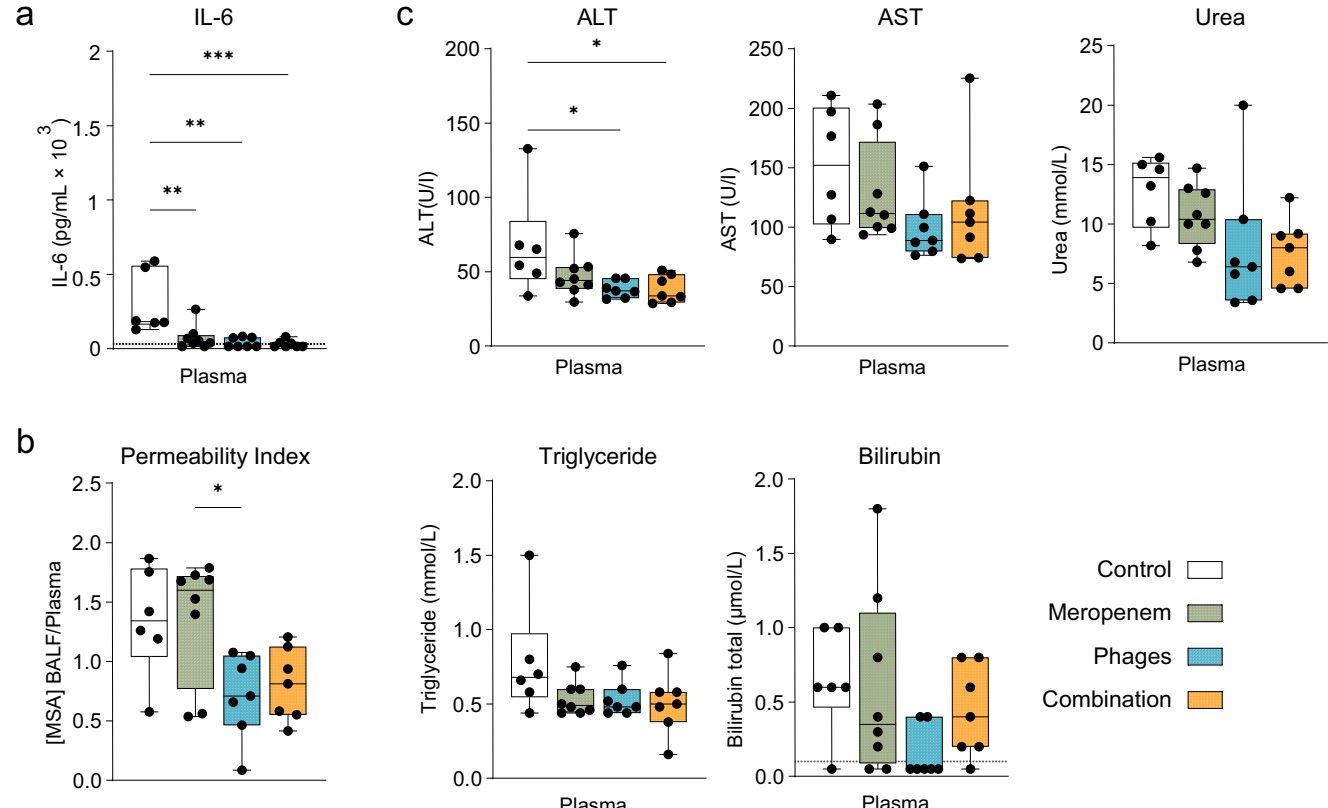

**Fig. 3 | Lung barrier function and systemic organ injury improved by phage treatment of mice with *Pseudomonas*-induced VAP. a** Plasma levels of IL-6, analyzed via IL-6 ELISA of mice with VAP at 24 hpi. **b** The lung permeability index was assessed by measuring the ratio of murine serum albumin (MSA) in BALF and plasma at 24 hpi. **c** Plasma levels of alanine aminotransferase (ALT), aspartate aminotransferase (AST), urea, triglyceride and total bilirubin reflecting the degree of organ injury in mice with VAP at 24 hpi, analyzed by Laboklin GmbH & Co. KG. Results are shown as box plots depicting median, quartiles, and range, as determined by ordinary one-way ANOVA with Tukey's multiple comparisons test: *$p < 0.05$, **$p < 0.01$, ***$p < 0.001$; $n = 6$–8 mice per group (Control group $n = 6$;

Meropenem group $n = 8$; Phages group $n = 7$; Combination group $n = 7$). Color coding for groups: Control group: white, Meropenem group: green, Phages group: blue; Combination group: orange. Dotted lines represent the detection limit. BALF bronchoalveolar lavage fluid, VAP ventilator-associated pneumonia. *P*-values were adjusted for multiple testing, exact *p*-values in order of appearance: **a** Control vs. Meropenem: $p = 0.0033$, Control vs. Phages: $p = 0.0012$, Control vs. Combination: $p = 0.0007$, **b** Permeability Index: Meropenem vs. Phages: $p = 0.0301$. **c** ALT: Control vs. Phages: $p = 0.0409$, Control vs. Combination: $p = 0.0412$. Source data are provided as a Source Data file.

## Adjunctive phage therapy prevents epithelial cell damage upon *Pseudomonas* infection

Next, we examined if the respective killing modes of phages and meropenem can inflict differential damage on epithelial cells and their barrier function beyond the effects of endotoxins, which we found not to differ significantly (see Fig. 5c). To this end, we measured the transepithelial electric resistance of epithelial cell monolayers upon exposure to supernatants derived from treated PAO1-LC (see Fig. 5) using the ECIS™ system, a well-established tool[49] (Fig. 6a). *Pseudomonas*-derived lipopolysaccharide (LPS) served as endotoxin control (Fig. 6b).

We observed significant differences in the normalized resistance over time, indicative of changes in monolayer integrity (Fig. 6b–e). In contrast to the phage-treated PAO1-LC, supernatants from the high-dose meropenem-treated PAO1-LC (50 × MIC) significantly reduced epithelial monolayer integrity in a time-dependent manner (Fig. 6c–f). The normalized resistance decreased over time up to -0.8 after 12 h, exceeding epithelial cell layer disruption provoked by high dose (5 μg/mL) LPS control (Fig. 6b). This LPS dose compares to -50 EU/mL endotoxins, 5-fold more than detected in the endotoxin quantification (see Fig. 5c), suggesting that the supernatants from meropenem-killed PAO1-LC contain bacterial cytotoxic factors beyond endotoxins.

In contrast, the supernatants of the phage-treated PAO1-LC (MOI 5) had no impact on the integrity of the cell layer regardless of the

sampling time point (Fig. 6d–f). The combination treatment (Combination ++++: Meropenem 5 × MIC, Phages MOI 0.5) showed minimal loss of cell integrity with supernatants from the 45 min time point (Fig. 6e). Supernatants collected at later time points showed no influence on cell monolayers; also, no bacteria were detectable (see Fig. 5b), reflecting the high antibacterial efficacy without adverse effects of the adjunctive phage treatment.

Having established that supernatant of PAO1-LC killed by meropenem harmed the integrity of epithelial cell monolayers, whereas those from phage-killed PAO1-LC did not (see Fig. 6), we next wondered if (a) mechanical disruption of *Pseudomonas* cultures would likewise inflict damage to the epithelial cells and (b) if the mediators of the damage are heat-sensitive molecules (Fig. 7).

Liquid bacterial cultures were disrupted by ultrasonication and supernatants before and after the lysis were used (see Figs. 6a, 7a). The filtered supernatants of PAO1-LC, sampled in the early logarithmic phase reflecting a highly virulent state of the bacteria before mechanical lysis, had no impact on the epithelial monolayers. Also, the supernatants after ultrasonication, regardless of filtration, showed no significant epithelial cell damage, as did the combination and phage-only-treated PAO1-LC (see Fig. 6d, e). Interestingly, heat-inactivation of the differentially treated supernatants reversed their barrier-altering properties (Fig. 7b). This indicates that simple disruption of bacteria does not suffice to release bacterial factors triggering epithelial

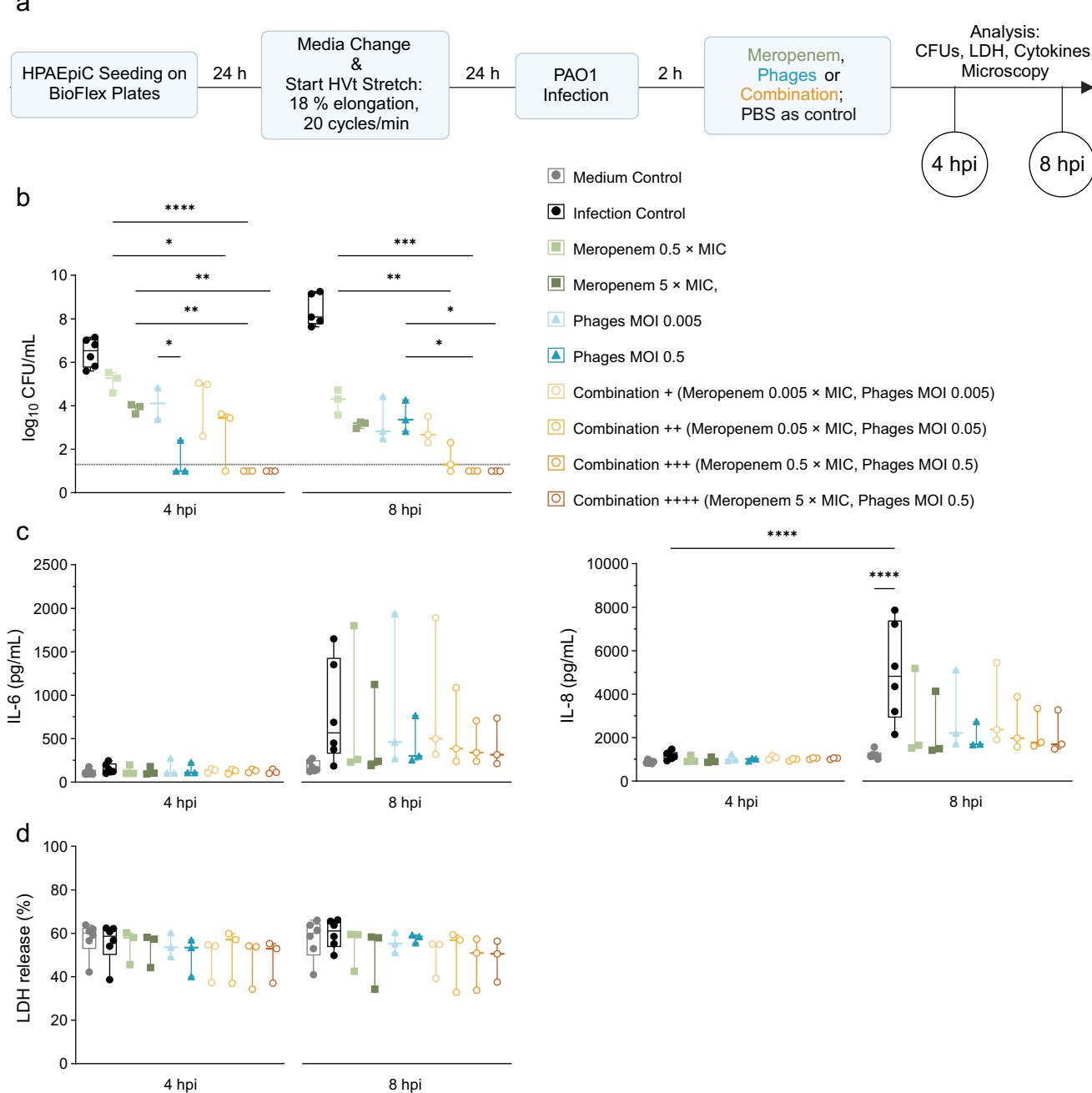

**Fig. 4 | Combination treatment of PAO1-infected HPAEpiC after 24 h high-tidal stretch most effective. a** Experimental plan created in BioRender. Nouailles, G. (2025) https://BioRender.com/padj3bg. HPAEpiC are seeded on 6-well BioFlex plates ($5 \times 10^5$ cells/well) and stretched for 24 h (18 % elongation; 20 stretch cycles/min) mimicking high tidal volume (HVt) mechanical ventilation, following PAO1 infection (MOI 1) and treatment with either meropenem (MIC = 1 μg/mL), the *Pseudomonas*-specific phage cocktail, or the combination of both with indicated concentrations. At indicated time points cell supernatants were analyzed for **b** CFUs, **c** inflammatory cytokines and **d** released LDH for cell death. **b** Logarithmic display (Y = log(CFU + 1)) of bacterial load at 4 hpi and 8 hpi. Levels of **c** IL-6, IL-8 and **d** LDH in collected supernatants, measured in duplicates according to manufacturer´s recommendations. LDH results reflecting cytotoxicity are expressed as percentage of total LDH release during cell lysis by 1% Triton X-100 for 4 h or 8 h, respectively. Data are shown as means of experiments with box plots depicting

median, quartiles, and range, as determined by two-way ANOVA with Tukey´s multiple comparisons test: *$p < 0.05$, **$p < 0.01$, ***$p < 0.001$, ****$p < 0.0001$; $n = 3$ independent experiments, while each condition was run in duplicates per experiment. Color and symbol coding for groups: Control group: circles and shades of gray, Meropenem group: squares and shades of green, Phages group: triangles and shades of blue; Combination group: open circles and shades of orange. Dotted line reflects detection limit. Cells without any infection or treatment serves as medium control, infected cells (MOI 1) without any treatment serves as infection control. CFUs Colony-forming units, HPAEpiC human primary alveolar epithelial cells, hpi hours post infection, IL Interleukin, LDH Lactate dehydrogenase, MIC minimum inhibitory concentration. *P*-values were adjusted for multiple testing, **b** exact *p*-values can be found in Supplementary Tables 2–4. Source data are provided as a Source Data file.

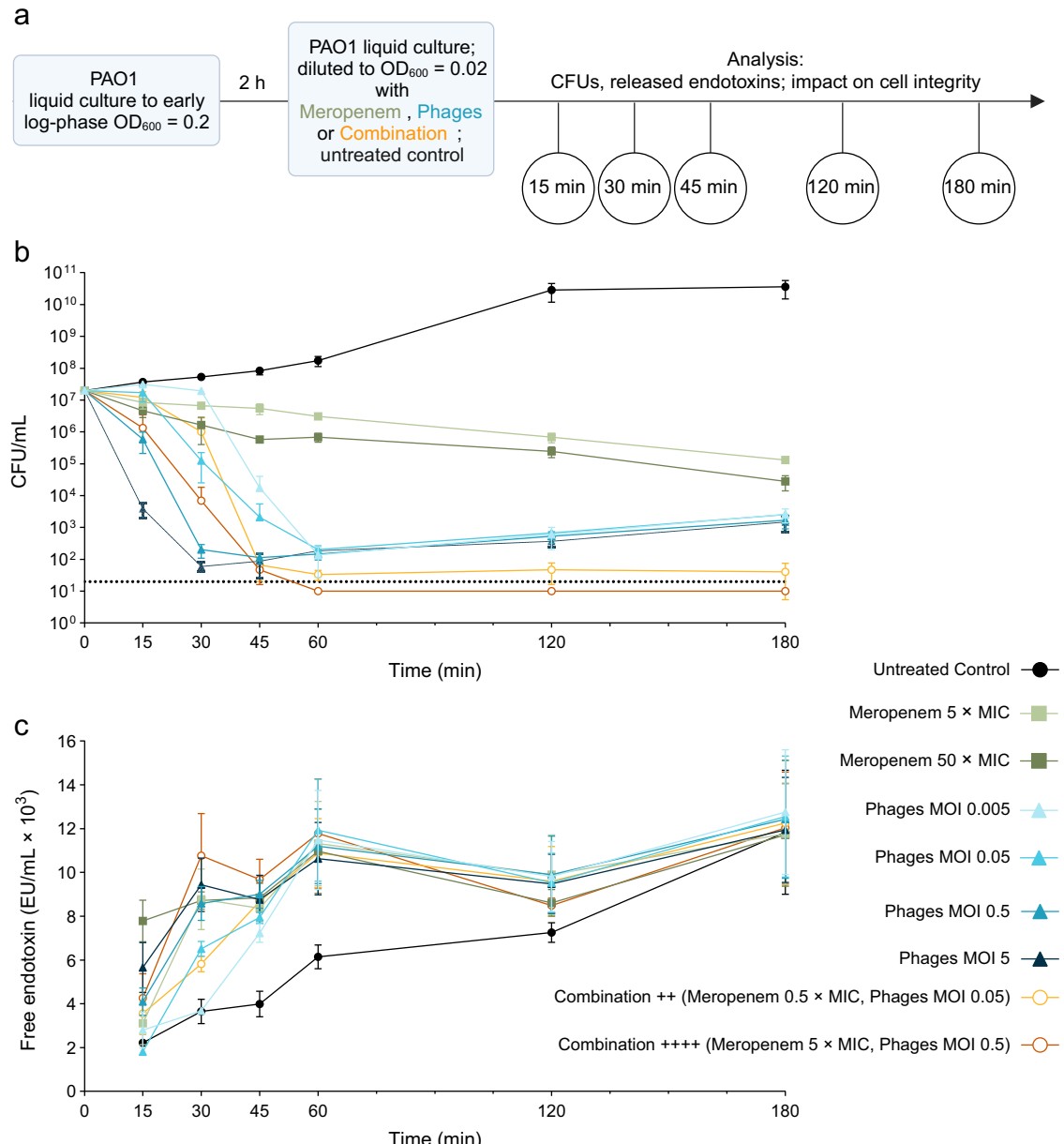

**Fig. 5 | Phage-antibiotic combination therapy is more effective in vitro than antibacterial agents alone. a** Experimental plan created in BioRender. Nouailles, G. (2025) https://BioRender.com/959k27u. For the infection assay PAO1 were cultured in the absence (control) or presence of meropenem (MIC = 1 μg/mL), the *Pseudomonas*-specific phage cocktail, or the combination of both in different concentrations as indicated. At different time points (15, 30, 45, 120, 180 min) liquid cultures were sampled (filtered 0.22 μm) and analyzed for bacteria load, released endotoxins and supernatants were used for experiments measuring epithelial monolayer integrity (see Figs. 6 and 7). **b** Colony-forming units (CFU/mL) according

to time. Data are expressed as means of 3 independent experiments ±SD. Color and symbol coding for groups: Control group: circles and shades of gray, Meropenem group: squares and shades of green, Phages group: triangles and shades of blue; Combination group: open circles and shades of orange. Dotted line reflects detection limit. **c** Concentration of free endotoxin (EU/mL) released in culture medium over time. Data are expressed as means of 3 independent experiments ±SEM. MIC minimum inhibitory concentration, MOI multiplicity of infection. Source data are provided as a Source Data file.

integrity loss, but rather that meropenem treatment specifically induces the release of cell-toxic bacterial components or virulence factors that are heat-sensitive, thus putatively proteinoid.

Finally, we infected the epithelial cell monolayers with PAO1 (see Fig. 6a) and compared the direct effect of the ongoing different mono-treatments with meropenem or the phage cocktail, or a combination of both (adjunctive phage therapy), on the barrier integrity over time for 72 h (Fig. 8, Supplementary Figs. S6–S8).

In line with the previous results, the adjunctive phage therapy was most effective in treating the bacterial infection and maintaining cell integrity (Fig. 8c, Supplementary Fig. S6c), reflecting the permeability

index of phage-treated mice with *Pseudomonas*-induced VAP in vivo (see Fig. 3b). The bacteria were effectively cleared at drastically lowered concentrations of the adjunctive treatment (Combination +: Meropenem 0.005 × MIC, Phages MOI 0.005) and the phage monotherapy (MOI 0.005) (Fig. 8b, Supplementary Fig. S6b). In contrast to the phage treatment groups, the effect of meropenem was dose-dependent, and the lowest dose of 0.05 × MIC failed to maintain cell integrity (Fig. 8a–d, Supplementary Fig. S6a). After 32 h, we observed complete disruption of the cell monolayer comparable to the infection control without any treatment, in which cell death occurred as early as 16 h. Without an infection, meropenem alone showed significant

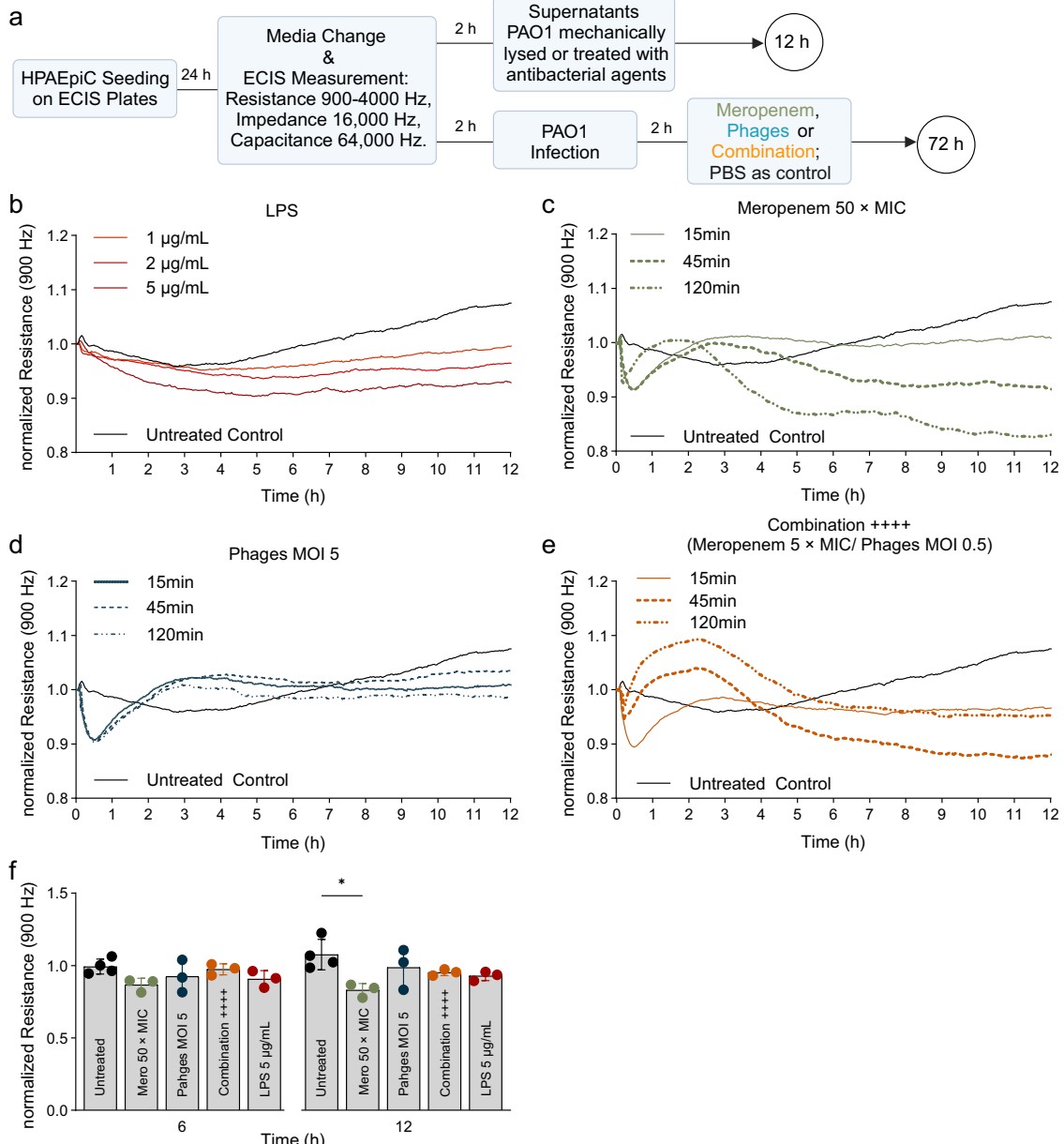

**Fig. 6 | Bacterial products after phage lysis did not affect epithelial cell integrity in vitro. a** Experimental plan created in BioRender. Nouailles, G. (2025) https://BioRender.com/wg3zy88. HPAEpiC monolayers in ECIS™ array wells were washed with antibiotic-free medium followed by (i) incubation with supernatants of the infection assay (see Fig. 5) or (ii) supernatants of mechanical lysed bacteria for 12 h (see Fig. 7); or (iii) followed by infection with PAO1 (MOI 1) and treatment with meropenem (MIC = 1 μg/mL), the *Pseudomonas*-specific phage cocktail, or the combination of both in indicated concentrations for 72 h (see Fig. 8). Cells incubated with different dosages of LPS (*P. aeruginosa* 10; Sigma-Aldrich) serves as positive controls for released endotoxins in the supernatants (-1 ng/mL LPS = 10 EU/mL endotoxins). Untreated Control are only HPAEpiC, same dataset in each figure. **b**–**e** Means of normalized resistance (900 Hz) versus time curves or **f** normalized resistance (900 Hz) of 120 min time point tested by two-way ANOVA with Tukey´s multiple comparisons test: *$p < 0.05$ are shown; $n = 3$–4 independent experiments, while each condition was run in triplicates per experiment. **b** $n_{5μg/ml} = 3$, all other groups $n = 4$ (**c**–**f**) $n_{Untreated\ control} = 4$, all other groups $n = 3$. **f** Data are displayed as mean ± SD. Color coding for groups: Control group: black, Meropenem group: shades of green, Phages group: shades of blue; Combination group: shades of orange; LPS group: shades of red. ECIS electric cell-substrate impedance sensing, EU endotoxin unit, HPAEpiC human primary alveolar epithelial cells, LPS lipopolysaccharide, MIC minimum inhibitory concentration, MOI multiplicity of infection. *P*-values were adjusted for multiple testing, exact *p*-values in order of appearance: **f** 12 hpi Untreated vs. Meropenem 50 × MIC: $p = 0.0102$. Source data are provided as a Source Data file.

epithelial cell damage, indicating potential cell-toxic effects of the antibiotic that were not observed to the same extent with phages alone (Supplementary Fig. S6a, b).

However, the direct comparison between the combination and the corresponding mono-treatments demonstrates the advantage of the adjunctive phage therapy (Combination ++: Meropenem 0.05 × MIC, Phages MOI 0.05) in maintaining cell integrity (Fig. 8d,

Supplementary Fig. S6d). The treatment with meropenem alone at a 0.05 × MIC concentration was ineffective. The phage monotherapy at a MOI of 0.05 showed only a minor reduction in cell integrity without complete cell death, indicating the phage cocktail's preventive antibacterial effect. Moreover, the lowest combination treatment (Combination +: Meropenem 0.005 × MIC, Phages MOI 0.005) demonstrated an effect comparable to the highest monotherapy with

meropenem (50 × MIC) or phages (MOI 50), further highlighting the robust potential of the adjunctive phage treatment strategy (Supplementary Fig. S7).

In conclusion, by combining the rapid control of bacterial growth by phages and the prolonged antibacterial activity of meropenem, the adjunctive phage therapy showed the best therapeutic outcome in vitro and in vivo, compared to monotherapies. Moreover, it offers the advantage of preventing the outgrowth of likely phage-resistant bacterial clones and maintaining lung barrier function.

## Discussion

Phage therapy has been proposed as an effective adjunctive treatment for infections caused by MDR-bacteria. Here, we show in a *Pseudomonas*-induced murine VAP model that combining the standard-of-care antibiotic meropenem with a *Pseudomonas*-specific phage cocktail accelerates disease amelioration, prevents lung and organ injury in vivo, and suppresses the outgrowth of likely phage-resistant bacteria in vitro, without increasing pro-inflammatory cytokine levels. Although meropenem monotherapy reduced bacterial levels similarly to adjunctive phage therapy, it failed to prevent the increase in inflammatory mediators or lung barrier damage. Our results suggest that the superior, longer-lasting efficacy of the adjunctive approach resulted from the rapid, specific bactericidal activity of phages paired with meropenems prolonged antibacterial effect, leading to faster and sustained bacterial clearance, reduced inflammation, and minimized tissue damage.

In accordance with the present results, increased antibiotic susceptibility due to phage-antibiotic combinations was also demonstrated by others: A recently published multicentre, multinational, retrospective observational study evidenced numerous examples of synergistic or additive interactions between phages and antibiotics, both in vivo and in vitro, leading to significant clinical improvements[50]. Treatment with an intravenous phage cocktail combined with systemic antibiotics from different classes successfully eradicated multidrug-resistant (MDR) *P. aeruginosa* pneumonia in a cystic fibrosis (CF) patient, restoring health and enabling lung transplantation[27]. Experimental studies demonstrated that combining phages and meropenem eradicated deep-dormant bacterial cultures in vitro and reduced persistent bacterial infections in mice[51] as well as improved survival in MDR-*Pseudomonas* pneumonia in immunocompromised mice compared to monotherapies[52].

Importantly, the regulation of host inflammation is crucial for preventing severe disease progression in pneumonia patients[53]. In our study, adjunctive phage treatment resulted in a significant reduction of pro-inflammatory cytokines (IL-6, IL-23, TNFα, IL-1α) in BALF, suggesting that this synergistic mechanism may ultimately lead to better outcomes in patients with VAP. For example, IL-23, which promotes neutrophil recruitment and pro-inflammatory gene expression in pulmonary epithelial cells via IL-17 signaling[54,55], was reduced following phage-only and combination treatments, as corroborated by a trend toward reduced IL-17 levels. Although neutrophil counts in BALF and lungs remained similar across all groups, phage-treated mice showed significant protection against lung barrier damage. Collectively, these results suggest that faster bacterial clearance by (adjunctive) phage therapy minimized inflammatory mediator release without affecting neutrophil recruitment.

To date, most experimental VAP studies have been conducted in pigs or piglets[43,56–58]. Accordingly, the efficacy of nebulized-phage therapy in *Pseudomonas*-induced VAP was first investigated in a pig model[21]. While prophylactic administration of nebulized phages improved survival in a rat model of VAP with methicillin-resistant *S. aureus* by reducing bacterial burden[59], combined antibiotic-phage therapy did not further enhance the therapeutic outcome[18] contrasting with our results. While large animal models offer several

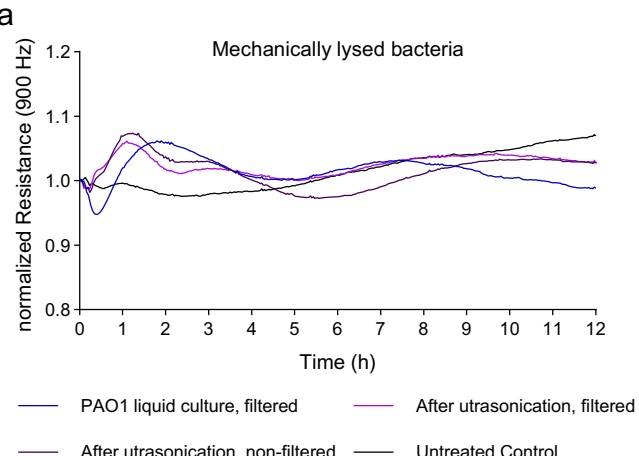

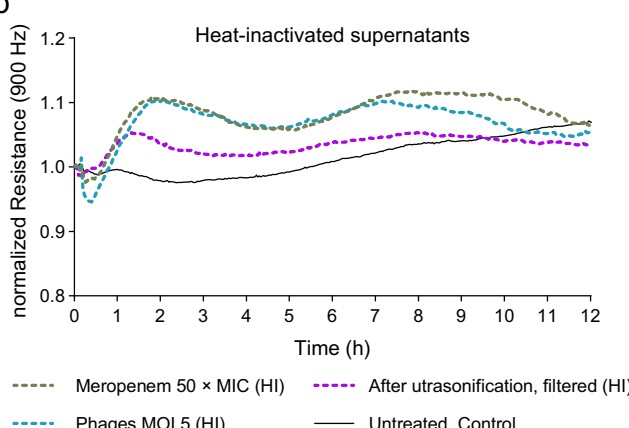

**Fig. 7 | Loss of epithelial integrity is prevented by heat inactivation of mechanically or antibacterially lysed *P. aeruginosa* cultures.** Measuring the transepithelial electric resistance (TEER) of epithelial cell monolayers by using ECIS™ system upon exposure to supernatants of **a** mechanical lysed bacteria via ultrasonication (Branson Sonifier 450, 400 W Titan-MicroTip) or **b** heat-inactivated (HI) supernatants of mechanically lysed or antibacterial treated PAO1-liquid cultures (120 min time point, see Fig. 5). Heat-inactivation was done at 60 °C for 60 min. Untreated Control are only HPAEpiC, same dataset in each figure. Means of normalized resistance (900 Hz) versus time curves are shown; *n* = 3. ECIS electric cell-substrate impedance sensing, HPAEpiC human primary alveolar epithelial cells, MIC minimum inhibitory concentration. Source data are provided as a Source Data file.

advantages in handling complex procedures (e. g. intubation procedures, ventilation time), our mouse model allows for a more in-depth characterization of the underlying immune response and associated lung barrier dysfunction. In conclusion, future clinical trials of adjunctive phage therapy should carefully consider the different experimental designs, when investigating variables such as phage selection, antibiotic choice, and their combination ratios.

In addition to the improved effectiveness of adjunctive phage therapy against *P. aeruginosa* compared to antibiotics alone, another advantage appears to be a delay in resistance emergence to either phages or antibiotics. In our study, in vitro experiments confirmed the synergistic efficacy of our adjunctive phage therapy in preventing the outgrowth of likely phage-resistant bacterial clones, one major issue of phage-only therapy[37]. Consistent with our study, in vitro treatment of *E. coli* with different combinations of antibiotics and phages demonstrated that sub-inhibitory concentrations of antibiotics during phage treatment control the emergence of phage-resistant mutants[41]. Using a

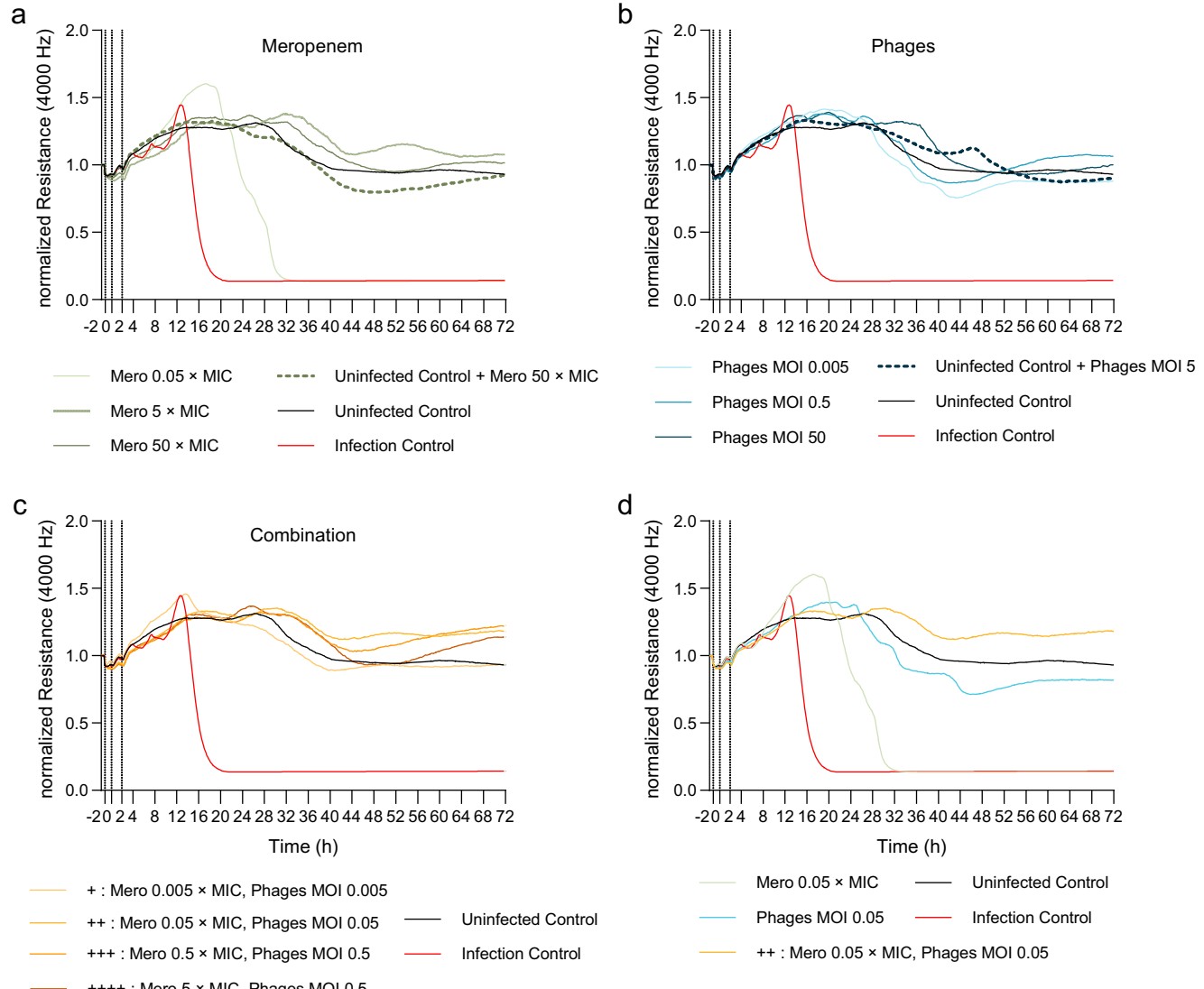

**Fig. 8 | Phage treatment improved lung epithelial cell integrity during *P. aeruginosa* infection in vitro.** Measuring the transepithelial electric resistance (TEER) of epithelial cell monolayers by using ECIS™ system upon infection with PAO1 (MOI 1) and subsequent treatment with **a** meropenem (Mero; MIC = 1 μg/mL), **b** the *Pseudomonas*-specific phage cocktail (Phages), or **c** a combination of both in indicated concentrations for 72 h. **d** Direct comparison of adjunctive phage therapy and single therapy. Means of normalized resistance (4000 Hz) versus time curves are shown; each isolate was run in 3 independent experiments while each condition where run in triplicates (96W10idf). Dotted line indicates the starting time of starving (−2 h), infection (0 h), and treatment (2 h). Uninfected Control and Infection Control (PAO1 MOI 1) same dataset in each figure. Uninfected Control are only HPAEpiC. Color coding for groups: Uninfected control group: black, Infected control group: red, Meropenem group: shades of green, Phages group: shades of blue; Combination group: shades of orange. ECIS electric cell-substrate impedance sensing, HPAEpiC human primary alveolar epithelial cells, MIC minimum inhibitory concentration. Source data are provided as a Source Data file.

cocktail of different phages generally results in a broader range of action targeting different bacterial strains and receptors, reducing the likelihood of resistant strains emerging during treatment[60,61]. Similar to our results, the efficacy of a personalized phage cocktail significantly enhanced therapeutic success, when screened against patients' bacterial isolates[25,26]. Additionally, more than 10-fold lower concentrations of meropenem were effective against *P. aeruginosa* infection in our study while maintaining lung epithelial integrity in vitro. Although the systematic assessment of phage- or antibiotic-resistant bacterial clone emergence still needs to be undertaken in the context of our murine VAP model, our results show high concordance with studies specifically addressing this aspect of adjunctive phage therapy[41,62].

Other studies have also reported that phage-antibiotic combination therapy can restore antibiotic susceptibility in resistant bacteria[63–65], providing a potential strategy for controlling highly virulent MDR-pathogens. A combination of phages and meropenem showed synergy against carbapenem-resistant *Acinetobacter baumannii* infection and its biofilm formation, as demonstrated by the authors in vitro, ex vivo, and in an in vivo zebrafish infection model[39]. This effect may be due to the promotion of phage proliferation and the recovery of meropenem susceptibility of newly phage-resistant bacteria. Furthermore, the development of phage-resistance can lead to reduced bacterial virulence[64,66,67]. Increased antibiotic susceptibility, enabling lower dosages or the use of alternative non-last-resort antibiotics, and preventing the development or reversal of bacterial antibiotic resistance through adjunctive phage therapy are strong arguments in favor of this approach. This strategy may be able to reduce the side effects of conventional antibiotic therapy while enhancing therapeutic effectiveness against MDR pathogens.

Notably, the mechanisms of antibacterial action for phages and meropenem differ: the lytic phages induce bacterial lysis at the end of their natural life cycle[17], while meropenem, a carbapenem antibiotic, exhibits high bactericidal activity by inhibition of bacterial cell wall synthesis[68]. The different modes of action may result in different amounts of free endotoxins, which are known to cause host cell damage. Antibacterial lysis of bacteria releases cell wall components, including endotoxins, which can activate the immune system, causing various inflammatory symptoms and pathophysiological disorders that harbor the risk of acute lung barrier damage and immediate adverse effects in critically ill patients[69,70]. In our study, phage treatment resulted in only a slightly delayed release of free endotoxins compared to meropenem alone in vitro. A study involving other β-lactams and two *E. coli* phages observed a more pronounced effect[71]. It is well established that different classes of antibiotics vary in their potential to release endotoxins; carbapenems, like meropenem, typically produce relatively low to intermediate amounts[70].

Despite the similar amounts of endotoxins released by both phages and meropenem in our experimental setup, only high-dose meropenem treatment significantly harmed epithelial cell monolayer integrity, while the presence of phages in the antibacterial treatment scheme prevented cell damage. *P. aeruginosa* produces a variety of virulence factors that can impair lung barrier function, mainly by intercellular junction disruption, leading to a loss of barrier integrity with severe clinical outcomes[72]. We found that meropenem treatment of liquid *P. aeruginosa* cultures explicitly induces the release of cell-toxic bacterial components or potential proteinoid virulence factors. However, the precise mode of action of how the components harm the epithelial barrier remains to be addressed in future studies.

A limitation of our study is the use of the laboratory *P. aeruginosa* strain PAO1 in our experimental murine model of *Pseudomonas*-induced VAP, which, although a relevant MDR pathogen, is not a clinical isolate. Including clinical isolates in the future could provide additional insights, but would require testing for suitable murine infection models and phage specificity. Moreover, the study is restricted to female mice to reduce variability in pneumonia severity introduced by sex[73–75]. Potential sex-specific effects need to be addressed in future studies. Additionally, assessing additional time points could offer a more comprehensive understanding of both innate and adaptive immune responses while enabling the tracking of resistant bacterial clones emerging over time in vivo. Further investigation is needed to validate our findings and determine whether bacterial mutants arise under adjunctive phage therapy or not, which would underscore the value of phage treatment as an adjuvant to conventional antibiotics. Unlike the intravenous (i. v.) administration of meropenem used in clinical settings, we chose intraperitoneal (i. p.) administrations in our study as both routes achieve comparable systemic distribution[76] while reducing handling time and experimental stress for the animals, thereby minimizing potential effects on disease severity or clinical improvement. Notably, since meropenem protein binding is five times higher in mice (20%) than in humans (2%)[76], the treatment doses required adjustment to reflect human dose regimens and ensure sustained antibiotic availability in the murine model.

In summary, adjunctive phage therapy holds significant potential for mechanically ventilated patients with ventilator-associated pneumonia (VAP). While our murine VAP study focused on *P. aeruginosa*, (adjunctive) phage therapy is also suitable for other MDR bacterial infections, such as *Klebsiella pneumoniae*, a major cause of both community-acquired and nosocomial infections[77,78]. In general, combination therapy offers key advantages over monotherapies, including the mitigation of excessive inflammation and associated lung barrier destabilization, as demonstrated in our study. Additionally, improved eradication of MDR bacteria, restoration of antibiotic sensitivity, and reduced bacterial virulence in phage-resistant strains offer further

advantages. These effects could translate into significant clinical benefits, potentially limiting acute lung injury while reducing the side effects of conventional antibacterial therapies and minimizing antibiotic overuse, thereby supporting a safer and more effective treatment option for future clinical applications.

## Methods

### Animals and ethics

All animal studies were approved by the institutional and local governmental authorities at the Charité–Universitätsmedizin Berlin and Landesamt für Gesundheit und Soziales (LaGeSo) Berlin. Mice were housed under specific pathogen-free conditions in individually ventilated cages with free access to food and water and a 12-hour (h) light/dark cycle, with an ambient temperature of 20–22 °C and humidity of 45–60%. Animal housing and experimental procedures complied with the Federation of European Laboratory Animal Science Associations (FELASA) guidelines and recommendations for the care and use of laboratory animals. Female 8- to 10-week-old C57BL/6J WT mice (Charles River, Sulzfeld, Germany) in randomly assigned groups were used. A pre-defined score sheet and a clinical disease score adapted from Wienhold et al.[19] (see Supplementary Table S1) based on specific murine pneumonia indicators were assessed prior to euthanasia; mice were monitored for body temperature (BAT-12 Microprobe, Physitemp, Clifton, NJ, USA), body weight and clinical signs of VAP at 4, 16, 24 h post-infection (hpi). Body temperature and body weight are represented in the clinical disease score and individually (Fig. 1) to facilitate cross-study comparisons.

### Bacterial strains and bacteriophages

For all experiments, the *P. aeruginosa* strain PAO1 (DSM 19880, DSMZ, German Collection of Microorganisms and Cell Cultures GmbH, Germany) was used. Bacteria cryostocks were streaked on Columbia agar plates containing 5% sheep blood (BD, Heidelberg, Germany) and incubated overnight at 37 °C with 5% $CO_2$. Subsequently, bacterial cultures were prepared from single colonies ($OD_{600} = 0.05–0.08$) in trypticase soy broth (TSB, Caso Bouillon, Carl Roth, Karlsruhe, Germany) and incubated at 37 °C and orbital shaking at 220 rpm until they reached early logarithmic phase ($OD_{600} = 0.2–0.3$). Notably, the PAO1 strain induces pneumonia independent of the inoculum's harvest growth phase[79,80]. After centrifugation ($1250 \times g$ for 10 min at room temperature (RT)), the pellet was resuspended in sterile Dulbecco's Phosphate-Buffered Saline (DPBS without Magnesium and Calcium, Thermo Fisher Scientific, Waltham, MA, USA) and adjusted to the desired inoculation dose for in vivo or in vitro experiments. Bacterial loads of tissue samples or cell culture supernatants were determined by plating serial dilutions in sterile $1 \times$ DPBS on blood agar plates (5% sheep blood) and incubating overnight prior to CFU counting. The detection limit was 20 CFU/mL.

In this study, an anti-*P. aeruginosa* phage cocktail was used, which includes two *Pseudomonas* phages: DSM 19872 (JG005)[44] and DSM 22045 (JG024)[45], kindly provided by the DSMZ (Braunschweig, Germany) and ITEM Fraunhofer (Braunschweig, Germany). The phages were previously selected against various *P. aeruginosa* strains by the providing institutions using the Routine Test Dilution method, as described elsewhere[81]. Additionally, the individual phage suspensions were highly purified through chromatography, and endotoxins were removed by the supplying institutions to low and non-immunogenic levels, as previously reported[31]. In a prior study[31], we demonstrated that intraperitoneal administration of the phages allowed them to reach the lungs, remain stable for several days, and remain viable. For the in vitro experiments a different batch of the same phages were used, in which the phages were mixed with 0.25% HSA to increase phage stability. The phage cocktail was assembled by mixing the single phage suspensions in phage buffer (0.1 M NaCl, 8 mM $MgSO_4$, 50 mM Tris-HCl, pH 7.2–7.5) immediately before each in vivo (~$5 \times 10^7$ plaque-

forming units (PFU)/phage per injection) or in vitro (indicated multiplicity of infection (MOI; ratio phages to bacteria) = 0.005–50) experiment to ensure accurate phage titers. The two phages are present in equal proportions within the phage cocktail. PFU analysis was performed with the strain PAO1. Phage titers of the original phage suspensions were confirmed with the corresponding indicator strains as reported previously[31]. For both phages, the lytic activity against the bacterial strain had been determined by the providing institutions previously and was confirmed in vitro via plaque-assays.

## Plaque assay

PFUs of tissue samples were determined by performing serial dilutions (1:10) in phage buffer shortly before the assay. Bacterial cultures were prepared (20 mL) from single colonies ($OD_{600} = 0.05–0.08$) and cultured with orbital shaking (220 rpm, at 37 °C) until they reached early logarithmic phase ($OD_{600} = 0.2–0.3$). Low melting agar (soft agar; 0.1% w/v technical agar BD, 5 mM $MgSO4$ Merck) (4 mL/glass tube) was melted in a heating block (110 °C for 10 min) and cooled down to 48 °C. For the spot test, 100 μL bacterial suspension was added to the liquid soft agar (top), gently mixed and poured on agar plates (bottom; 1.5% w/v technical agar, BD). Subsequently, sample dilutions were spotted (4 μL/spot) in triplicate on the plates. Agar plates were incubated overnight (37 °C, 5% $CO_2$) before calculating phage load per tissue sample. The detection limit of the triplicates from the spot test was 83 PFU/mL.

## Murine ventilator-associated pneumonia (VAP) model

Mice were anaesthetized by intraperitoneal (i. p.) injection (0.08 mg/kg fentanyl, 8 mg/kg midazolam, and 0.8 mg/kg medetomidine), orotracheally intubated and mechanical ventilated (MV; FlexiVent, SCIREQ, Montreal, Canada) for 4 h with high tidal volume (HVt: 34 mL/kg body weight)[46,47]. The establishment of the murine *Pseudomonas*-induced VAP model was previously conducted and detailed in our previous work[47], including procedure control groups to assess the effects of the ventilation versus infection. Following an initial recruitment maneuver, the lung function parameters such as mean airway pressure, dynamic elastance, as well as static compliance and inspiratory capacity were recorded at 20 min intervals during MV. The terminal recruitment maneuver was performed 5 min prior to termination of MV. After 4 h of MV, mice were removed from the ventilator and 20 μl bacterial suspension containing $5 \times 10^4$ CFUs of PAO1 in 20 μL sterile 1 × DPBS was instilled via the endotracheal tube. For post-anesthesia recovery, mice were injected with 0.5 mg/kg flumazenil and 2.5 mg/kg atipamezole to antagonize the narcosis, followed by extubation and spontaneous breathing for 24 h.

At 4 and 16 hpi, mice were treated by i. p. injections with 100 μL of either meropenem (Dr. Eberth, PZN 10170625), the anti-*P. aeruginosa* phage cocktail (~$5 \times 10^7$ PFU/injection per phage), or a combination of both. The antibiotic solutions were prepared from intravenous formulations used for humans, reconstituted with sterile NaCl (0.9%; B. Braun, Melsungen, Germany) to a final concentration of 10 mg/mouse per injection (~1 g/kg body weight/day based on published PK/PD correlation analysis[76]). Sterile NaCl (0.9%; B. Braun, Melsungen, Germany) served as control group.

At 24 hpi, mice were euthanized with ketamine (200 mg/kg body weight) and xylazine (20 mg/kg body weight) and sampled for blood, bronchoalveolar lavage (BAL), lungs, liver, and spleen for further analysis. After exsanguination via the vena cava, the tracheas were cannulated and lungs lavaged twice with 0.8 mL 1 × DPBS protease inhibitor (PI) solution (cOmplete™, Mini Protease Inhibitor Cocktail; Roche, Basel, Switzerland). Lavaged lungs were perfused via the right heart chamber with 10 mL 1 × DPBS, divided equally, and homogenized in 1 mL 1 × DPBS PI-solution with gentleMACS™ M tubes (Miltenyi Biotec, Bergisch Gladbach, Germany) or minced and digested for cell isolation. Spleens and livers were removed and homogenized in

1–3 mL 1 × DPBS PI-solution using gentleMACS™ M tubes. Whole blood in EDTA tubes (Sarstedt, Nümbrecht, Germany) was centrifuged at 1500 × g for 10 min at 4 °C, plasma was frozen in liquid nitrogen and stored at −80 °C until further analysis.

Treatment efficacy and lung injury were quantified by lung function parameters, pulmonary permeability, flow cytometry analysis of innate immune cells, and inflammatory markers. Bacterial load was measured in BAL, homogenized organs (lungs, spleen, liver) and whole blood as described above. The determination of phage load was done using supernatants of centrifuged homogenized organs (1250 × g, 10 min, 4 °C) and whole blood as described above.

## Leukocyte differentiation in BAL and lungs by flow cytometry

Innate immune cells (leukocytes) in BAL and lung homogenate were analyzed by flow cytometry (BD FACSCanto™ II, BD, Heidelberg, Germany) as described previously[82]. For the detailed gating strategy refer to Fig. S1 (Supplementary Materials). In brief, minced lungs were digested in RPMI media containing collagenase II (Biochrome, Berlin, Germany) and DNAse I (PanReac AppliChem, Darmstadt, Germany) for 30 min at 37 °C and a single-cell suspension prepared using a 70 μm cell strainer (BD, Heidelberg, Germany). The suspension was centrifuged (470 × g, 5 min, 4 °C) and the pellet resuspended in 2 mL 1 × red blood cell lysis buffer (Santa Cruz Biotechnology, Dallas, USA) for 2 min. Cells were washed and centrifuged, cell pellets resuspended in PBS/0.1% BSA buffer and analyzed by flow cytometry.

For flow cytometry analysis, BAL and lung cells were blocked with anti-CD16/CD32 (2.4G2; cat. nr. 553142; BD, Heidelberg, Germany) and stained with anti-CD45 (30F11; cat. nr. 103108; BD, Heidelberg, Germany), anti-CD11c (HL3; cat. nr. 550261; BD, Heidelberg, Germany), anti-CD11b (M1/70; cat. nr. 25-0112-82; eBioscience, Frankfurt, Germany), anti-F4/80 (BM8; cat. nr. 12-4801-80; eBioscience, Frankfurt, Germany), anti-Ly6G (1A8; cat. nr. 560602; BD, Heidelberg, Germany), anti-LyGC (HK.1.4; cat. nr. 128033; BioLegend, San Diego, USA), anti-MHCII (M5/114.15.2; cat. nr. 56-5321-82; eBioscience, Frankfurt, Germany), or anti-Siglec F (E502440; cat. nr. 562681; BD, Heidelberg, Germany) monoclonal antibodies (mAbs). CountBright Absolute Counting Beads (Thermo Fisher Scientific, Waltham, USA) were used for calculation of total cell numbers.

## Clinical blood analysis

The degree of organ injury in treated mice with VAP was assessed by quantifying plasma levels of aspartate aminotransferase (AST), alanine aminotransferase (ALT), urea, triglyceride and total bilirubin. Plasma samples were analyzed by LABOKLIN GmbH & Co. KG (Bad Kissingen, Germany).

## Alveolar-capillary barrier permeability index

To quantify pulmonary vascular leakage, the alveolar-capillary barrier permeability index was assessed by measuring mouse serum albumin (MSA) levels in BAL fluid (BALF) and plasma by ELISA (Mouse Albumin ELISA Quantification Set, BETHYL Laboratories, Montgomery, TX, USA) according to the manufacturer's instructions and analyzed via SkanIt Software for Microplate Readers RE, ver 7.0.1.4. The BALF/plasma MSA ratio was calculated as the permeability index of the alveolar-capillary barrier as described elsewhere[82].

## Cytokine and chemokine quantification

Inflammatory cytokines and chemokines in murine plasma and BALF samples were measured using the Mouse Inflammation Panel (13-plex with V-bottom plates; BioLegend, San Diego, CA, USA) according to the manufacturer's recommendations and the corresponding software provided by BioLegend.

Soluble cytokines IL-6 and IL-8 in cell culture supernatants were measured by BD OptEIA™ ELISA kits (BD Biosciences) according to

manufacturer's recommendations and analyzed via SkanIt Software for Microplate Readers RE, ver 7.0.1.4.

## Lactate dehydrogenase assay

Lactate dehydrogenase (LDH) is released during cell lysis and used as an alternative marker for cell death. Levels of LDH in cell culture medium were measured using the Cytotoxicity Detection Kit (Roche, Basel, Switzerland) according to the manufacturer's recommendations and analyzed via SkanIt Software for Microplate Readers RE, ver 7.0.1.4. Data are expressed as percentage of total LDH release during cell lysis by 1% Triton X-100.

## Infection assay and endotoxin quantification

For the infection assay, a liquid culture of PAO1 was prepared as described above until reaching early logarithmic phase ($OD_{600} = 0.2$). The liquid culture was diluted with fresh medium to an $OD_{600} = 0.02$ ($2 \times 10^7$ CFU/mL) in 10 mL, mixed with meropenem (minimum inhibitory concentration; MIC = 1 μg/mL[83–85]), the anti-*P. aeruginosa* phage cocktail, or the combination of both in indicated concentrations and incubated at 37 °C and orbital shaking at 220 rpm. At different time points (15, 30, 45, 60, 120, 180 min) the individual approaches were first sampled for calculating the CFU/mL, then sterile filtered (0.22 μm filter; Carl Roth, Karlsruhe, Germany) and stored at −20 °C until further analysis.

Free endotoxin concentrations were quantified using the Pierce™ Chromogenic Endotoxin Quant Kit (Thermo Fisher Scientific, Waltham, MA, USA), an amebocyte lysate assay, after a 1:1000 dilution and according to the manufacturer instructions and analyzed via SkanIt Software for Microplate Readers RE, ver 7.0.1.4.

## Mechanical disruption of liquid cultures of PAO1 via ultrasonication

Liquid cultures of PAO1 were prepared as described above until they reached early logarithmic phase ($OD_{600} = 0.2$). Cell pellets after centrifugation ($1250 \times g$, 10 min, RT) were washed once in sterile $1 \times$ DPBS, resuspended in 5 mL $1 \times$ DPBS, and handled on ice. Mechanical lysis of bacterial cultures was carried out in cooperation with Dr. Robert Hurwitz (MPI Berlin, Germany) via ultrasonification using a Branson Sonifier 450, 400 W Titan-Microtip in $4 \times 10$ cycles (output control: 4, duty cycle: 40%) per culture. Lysed bacterial suspension or residual supernatants after centrifugation ($1250 \times g$, 10 min, RT) were filtered (0.22 μm). Samples of bacterial suspension before and after the ultrasonication, as well as before and after filtration were stored at 4 °C until further use.

Heat-inactivation of bacterial cultures, lysed suspensions or supernatants were done for 60 min at 60 °C.

## Cell culture

Human primary alveolar epithelial cells (HPAEpiC) from human lung tissue were purchased from Cell Biologics (H-6053; Chicago, Illinois, USA). Cells were cultured in T75 flasks (Sarstedt, Nümbrecht, Germany) as per the manufacturer´s recommendations with Human Epithelial Cell Medium (Cell biologics, H6621) with supplements and antibiotics (Cell Biologics, Chicago, USA; Supplement growth Kit H6621-Kit: 0.1% endothelial-growth factor (EGF), 0.1% hydrocortisone, 1% antibiotic-antimycotic solution, 5% fetal bovine serum (FBS) at 37 °C, 5% $CO_2$.

## Mechanical stretch of human primary epithelial cells

To mimic acute respiratory distress syndrome (ARDS) in vitro we used the Flexcell® FX-5000™ Tension System (Flexcell International, Hillsborough, NC). HPAEpiC were seeded ($5 \times 10^5$ cells/well) 1 day prior to stretch on pre-warmed BioFlex culture plates coated with collagen

type IV (Flexcell International, Hillsborough, NC). Directly before starting the stretching protocol, the medium was changed to antibiotic-free medium to obtain an adequate infection afterwards. To mimic HVt ventilation cells were stretched with 18% elongation and 20 stretch cycles/min for 24 h[47]. After the stretch, cells were immediately infected with PAO1 (MOI 1; ratio bacteria to cells: $5 \times 10^5$ CFUs/well), incubated for 2 h and treated with meropenem (MIC = 1 μg/mL[83–85]), the anti-*P. aeruginosa* phage cocktail or a combination of both at indicated concentrations at 2 hpi. At 4 hpi and 8 hpi, cell supernatants were first collected for determination of bacterial loads, then centrifuged (10 min, $1500 \times g$, 4 °C) and stored at −20 °C until further analysis of levels of LDH and cytokines.

## Electric cell substrate impedance sensing (ECIS) measurements

For assessing the lung barrier function under phage treatment in vitro, the ECISTM Z-Theta system (Applied Biophysics, Inc. Troy, NY, USA) with the provided Applied BioPhysics–ECIS Software v1.2.254.0. PC was used to measure impedance, resistance, and capacitance. ECIS electrodes (96W10idf) were placed in a holder plate in a humid incubator at 37 °C and 5% $CO_2$. In this study, the following default optimal frequencies were used: resistance (R) 900/4000 Hz, impedance (Z) 16,000 Hz, and capacitance (C) 64,000 Hz. Before cell seeding, the arrays were coated with gelatin solution (Cell Biologics, Chicago, USA) for 15 min at 37 °C for HPAEpiC (HPAEpiC complete medium; Cell Biologics, Chicago, USA). Coating solution was removed and cell medium was added for baseline measurements for 1 h. Following stabilization, the array was removed from the array station, cells were seeded (96W10idf: $5 \times 10^4$ cells/200 μL) and baseline measurements were continued up to 48 h. Arrays were washed with antibiotic-free medium for 1–2 h followed by (i) incubation with different supernatants (diluted 1:3) from the infection assay, (ii) supernatants of mechanical lysed bacteria or (iii) infection with PAO1 (MOI 1) and subsequent treatment with meropenem (MIC = 1 μg/mL[83–85]), the anti-*P. aeruginosa* phage cocktail or a combination of both at indicated concentrations. Cells incubated with different dosages of lipopolysaccharide (LPS, *P. aeruginosa* 10; Sigma-Aldrich, St. Louis, Missouri, USA) served as positive controls for released endotoxins in the supernatants and cells without treatment served as untreated/uninfected controls. Real-time measurements were initiated for 12–72 h after cell treatment. Data are expressed as normalized resistance (900 Hz; 4000 Hz) versus time curves; each isolate was run in 3–4 independent experiments while each condition where run in triplicates.

## Statistical analysis

Data were analyzed using GraphPad Prism 10 (San Diego, CA, USA). For grouped analyses, ordinary one-way or two-way analysis of variance (ANOVA) with Tukey's multiple comparisons test was performed. Results were considered significant if P was less than 0.05. Significance levels are indicated in the figures, sample sizes of individual groups are indicated in the figure legends. CFU and PFU data were logarithmized ($Y = \log(CFU + 1)$) when indicated. Detection limits are indicated in the figures.

## Reporting summary

Further information on research design is available in the Nature Portfolio Reporting Summary linked to this article.

# Data availability

Source data are provided with this paper. The experimental murine in vivo and human in vitro data generated in this study are provided in the Supplementary Information and in the Source Data file. Source data are provided with this paper.

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

## Acknowledgements

The authors would like to thank the Fraunhofer Institute for Toxicology and Experimental Medicine, especially S. Wienecke and H. Ziehr, for purified phage preparation and S. Häußler (TWINCORE, Centre for Experimental and Clinical Infection Research, Hannover, Germany and Department of Clinical Microbiology, Copenhagen University Hospital-Rigshospitalet, Copenhagen, Denmark) for providing *P. aeruginosa* clinical strains. During the preparation of this work the authors used ChatGPT / OpenAI, Grammarly / Grammarly, Inc. and DeepL / DeepL SE in order to improve English language and grammar, as well as German to English translations. After using these tools, the authors reviewed and edited the content as needed and take full responsibility for the content of the publication. This work received funding from Bundesministerium für Bildung und Forschung (BMBF), Germany and the Agence Nationale de la Recherche, France under the collaborative grant "MAPVAP", FKZ 01KI2124 (G.N., M.W.) and ANR-19-AMRB-0002 (J.-D.R., L.D.). This work was supported by Deutsche Forschungsgemeinschaft (DFG) grant SFB TR84 (M.W.); Deutsche Forschungsgemeinschaft grant SFB 1449–431232613 (M.W. and G.N.); BMBF e:Med CAPSyS grant 01ZX1604B (M.W.); BMBF e:Med SYMPATH grant 01ZX1906A (M.W.). The founders had no role in the design of the study, in the collection, analyses, or interpretation of data, in the writing of the manuscript, or in the decision to publish the results.

## Author contributions

Conceptualization: C.W., G.N.; Methodology: C.W., J.L., U.B., K.H., M.B., C.T., G.K., B.G., L.D., M.F., G.N.; Data Acquisition: C.W., J.L., U.B., K.H.; Data interpretation: C.W., J.L., K.H., M.B., G.K., B.G., L.D., J.-D.R., M.W., M.F., G.N.; Ressources: I.H.E.K, C.R., L.D., J.-D.R., M.W., M.F., G.N.; Supervision: G.K., C.R., L.D., J.-D.R., M.W., M.F., G.N.; Funding acquisition: M.W., G.N.; Writing—original draft: C.W., M.F., G.N.; Writing—revision and editing: all authors. All authors have read and agreed to the submitted version of the manuscript.

## Funding

## Competing interests

M.W. received funding for research from Biotest AG, Boehringer Ingelheim, Pantherna, Vaxxilon, and for talks or advisory from Actelion, Aptarion, Astra Zeneca, Bayer Health Care, Berlin Chemie, Biotest, Boehringer Ingelheim, Chiesi, Glaxo Smith Kline, Novartis, Pantherna, Teva. G.N. received funding for research from Biotest AG. The remaining authors declare no competing interests.
