## [Transparent Peer Review file · Nature Communications]

Adjunctive phage therapy improves antibiotic treatment of Ventilator-Associated-Pneumonia with *Pseudomonas aeruginosa*

Corresponding Author: Ms Chantal Weissfuss

Version 0:

Reviewer comments:

Reviewer #1

(Remarks to the Author)

Chantal Weissfuss et al. evaluated the combination of a two-phage cocktail with carbapenem antibiotic meropenem for the treatment of ventilator-associated *P. aeruginosa* pneumonia in mice. They demonstrated that addition of phage therapy improved the clinical recovery of meropenem treated animals, without increasing pro-inflammatory cytokine release. Addition of meropenem on the other hand prevented the emergence of phage-resistant clones. The authors uses unique methods to compare the effects of the various therapies on lung physiology features, such as lung barrier permeability index and function, transepithelial electric resistance, using specific settings, including stretched human primary epithelial cell cultures to mimic ARDS. They observed that adjunctive phage therapy prevents epithelial damage upon *P. aeruginosa* infection and improved lung epithelial cell integrity when combined to meropenem.

The evaluation of phage and antibiotics in experimental models of pneumonia in rodents is not original per se, nor is the evaluation of the host immune response under phage therapy, yet the evaluation of the effect of those therapies on lung epithelia is quite original. However, several major concerns on the methodology of the experiments tempered my enthusiasm.

Here are the most important ones.

EXPERIMENTAL PNEUMONIA MODEL

1) in a 2010 paper from the same authors, the inoculum chosen was much higher (1×10^7) which resulted in much higher bacterial loads in BAL (1.6×10^8 versus ca. 10^4 , Fig.1f). A dose-finding study justifying for the inoculum choice is thus missing.

2) a PK study of phages in BAL after i.p. injection is missing (phage without infection group). This group is essential to compute the expected MOIs, to prove that phages underwent a virulent cycle and not that the decrease in bacterial load observed is due to lysis from without.

3) A control group - anesthesia/ventilation only - is missing to assess what is due to the procedure and what is due to the infection in the clinical outcome.

4) Why were the animals killed after 24h? Why was survival not evaluated as secondary outcome? For instance in the previous 2010 study, the animals were followed over 72h. This is absolutely needed to confirm the relevance of the observations.

5) The meropenem dose (1g/kg/d) is clinically irrelevant - for a human being max recommended schemes are around 40 mg/kg/tid = 120 mg/kg/d (or normally 3x1g, max 3x2g for an adult of ca.70-80kg). Authors did not mention the MIC of PAO1 to meropenem either.

6) The way the inocula were prepared might not allow for correct assessment of *P. aeruginosa* virulence. Authors grew isolates up to early exponential phases a time of growth when not all Quorum sensing systems are active (mid exponential

phases would have been better). As sign for it, the level of inflammatory cytokines measured are lower compared to the 2010 study (IL-6). As host response is one of the read-out the authors assessed, it is essential to look for a virulent bacterial infection, if not the results on the decreased inflammation induced by phages might well be irrelevant.

7) The differences in clinical scores between meropenem and phages observed at 16h is slowly vanishing at 24h, which mandates a prolonged observation time, as beta-lactams need some time to be effective (see below comment on time kill curves).

8) PFU assessment in tissue samples in parallel to CFU is missing. My question would be the following: does meropenem reduce the phage replication by decreasing the number of hosts available?

TIME-KILL CURVES (Fig.5)

1) I questioned the rationale of the time-line chosen for this experiment, which cannot assess meropenem efficacy. Indeed, time over MIC is the PK/PD index to be targeted for beta-lactam efficacy. Ending the experiment after 3 hours is not enough for meropenem to be efficient, as demonstrated by the only minor reduction of the bacterial load. It is well known that bactericidal effect of phages is fast, but the way the results are presented over such short period of time might be misleading for the comparison with meropenem.

2) I would use multiple of MIC rather than meropenem concentrations in the legend.

3) meropenem concentrations as high as 50 mg/L are not clinically relevant. Usual through target levels are supposed to be around 5-10 mg/L. one might observe peak around 50 mg/L but as continuous/prolonged infusion is being currently recommended, such concentrations are becoming less common.

4) An observation time beyond 3 hours might have highlighted the emergence of phage resistant mutants. When this phenomenon was observed, were those mutants analyzed (Fig4b)?

In summary, while authors looked at lung cell injury using advanced assays, there are several issues in the experimental design, such as short observation time, missing control groups (e.g. phage only, procedure only), Dose-finding/PK studies, PFU assessment for animal studies, which questioned the relevance of the results. In addition, it is not clear which therapy is adjunctive to which one, which makes the reading sometimes confusing (e.g. line 426-428 the adjunctive phage therapy is not preventing the outgrowth of phage-resistant clones, but rather meropenem). Authors should make clear, what they considered standard of care, what they consider as adjunctive therapy, etc.

Reviewer #2

(Remarks to the Author)

The manuscript by Weissfuss et al describes a comprehensive study of phage and antibiotic therapy against *P. aeruginosa* in a VAP model, including an initial in vivo study followed by extensive in vitro studies. This is a useful study that benefits from a tight focus on a phage combination and one antibiotic, which allows for more depth of study. In clinical use, phage are often used alongside standard antibiotics and the interaction of these two agents is important for development of phages as therapeutics. The manuscript is generally well-written and well-presented, with relatively minor revisions needed. One issue that must be addressed is a better description of how the phages were prepared for in vivo and tissue culture experiments, noted below. The discussion section is very comprehensive but rather lengthy and could also benefit from some condensing.

L 41: Ventilator-associated pneumonia

L47: in recent years

L 50: selecting for carbapenem-resistant pathogens

L 117-124: how were the phages prepared for in vivo and tissue culture use, were they purified to remove cell debris, media and bacterial endotoxin? Unpurified phage lysates usually contain large amounts of endotoxin (thousands of EU/ml) that can cause inflammatory responses depending on how they are administered.

L 334-336: this is not what is shown in the figure, there is no statistical significance in the differences between groups.

L 454, 455: use "to" instead of "up to" here.

L 457-459: is it known if the two phages use the same receptor, or if they exhibit cross-resistance (i.e., bacterial mutation to resistance against one phage also makes them resistant to the other phage)?

L 497: exceeding

L 502: In contrast,

Fig. 1: clinical score includes temperature and weight, so it seems like these factors are being counted twice here.

Fig. 4: indicate what infection control and medium control mean here

Reviewer #3

(Remarks to the Author)

"Adjunctive phage therapy improves antibiotic treatment of Ventilator-Associated-Pneumonia with *Pseudomonas*

aeruginosa" (NCOMMS-24-52003) by Weissfuss et. al.

Comments for Author

Weissfuss et al. aimed to investigate the effectiveness of combining antibiotics, specifically, meropenem with phages to improve their effectiveness in a model of ventilator-associated pneumonia (VAP), induced by the Gram-negative bacterium *Pseudomonas aeruginosa*.

The authors convincingly demonstrate the enhanced effectiveness of antibiotic-phage combination therapy over that of traditional antibiotic monotherapy for the treatment of VAP. Specifically, in a VAP mouse model, in initial experiments they demonstrate that mice receiving combination therapy have a faster clinical recovery, compared to those of monotherapy, albeit no differences in lung bacterial burden are evident. They further show that local inflammatory mediator (IL-6), as expected, increased in the VAP model, but is significantly reduced following phage therapy alone, or in combination with meropenem.

They further demonstrate that lung barrier function and systemic organ injury is improved by phage treatment in VAP.

These in vivo findings were corroborated in vitro, whereby the authors demonstrate a dose dependent reduction in bacteria using combination therapy, which displayed the most significant bacterial killing capacity at 5ug/ml meropenem; lower than the usual minimum effective dose of 50ug/ml. These studies were also complemented by in vitro human data where it is demonstrated that combination therapy prevents bacterial-induced damage to human primary epithelial cells.

Taken together, the study by Weissfuss et. al. clearly demonstrates that adjunctive phage therapy improves antibiotic treatment during *P. aeruginosa*-induced VAP. The study and methodological approaches appear to be technically sound and in sufficient detail to be reproduced. The manuscript includes high-quality data that will be of interest to microbiologists, those studying host-pathogen interactions, and antimicrobial resistance, in addition to related fields including airway innate immunity and antibiotic drug delivery approaches. The authors present data to support an alternative therapeutic approach to acute VAP that has improved efficacy and has fewer adverse effects. Given the increased acquisition of multidrug resistance strains, there is an urgent and unmet need for novel treatment options for critically ill patients' development VAP. Accordingly, this research is important and timely. Despite this, the mechanism behind the synergistic effects of combination therapy presented herein is currently unknown. The use of clinical strains, rather than the laboratory strain PAO1 would have significantly improved the manuscript. For example, performing selected key experiments, similar to those in Fig 1 and 2 with a clinical isolate to demonstrate clinical relevance/effectiveness, would have been greatly beneficial. However, the authors do acknowledge this as a limitation to the study and the main aim to assess if adjunctive therapy could improve antibiotic treatment during VAP has been achieved.

Major:

Meropenem alone and combination reduce bacterial burden to a similar level (no difference between the two groups), yet combination said to be more effective, what defines recovery?

Figure 2 – The inclusion of BALF cytokines, aside from IL-6 would be beneficial, particularly those which change in the VAP model and specifically with phage therapy – IL23

Line 149 – female mice are used for all experiments, but this is not justified. Male and Female mice have differing levels of susceptibility to respiratory bacterial infection and as such, justification as to why only female mice are used throughout this study is warranted.

Line 167 – meropenem is usually administrated i.v. Herein, i.p injections are used. Justification as to why this approach is used and how it translates clinically is required.

Line 413 – Figure 4. do dot plots represent the mean of independent experiments, or as stated technical duplicates? If so, is this data from one single experiment? N numbers should be clarified.

Figure 5 b and c – these figures are quite busy, with multiple groups displayed on the graphs. Graphs could be reformatted or enlarged to improve clarify for the reader

Line 490. Figure 6 – As per Figure 4, it is unclear how many independent experiments were performed. This should be clarified throughout all figures. Data presented as the mean of independent experiments or are these technical duplicates?

Line 363 & Line 345 – "As we observed good results as the primary – site of infection" – No significant differences in bacterial burden, one of the main readouts of infection status, was evident in the combined therapy group when compared to individual therapy (meropenem alone). Concurrently, no significant differences in PMN or leukocytes are evident in either treatment groups (meropenem alone, versus combination).

Line 245 – "mice that received adjunctive phage therapy benefited the most, with improved clinical scores and lowest bacterial burden compared to either meropenem or phage cocktail therapy." While it is acknowledged that combine therapy resulted in improved clinical scores, there were no significant differences in bacterial burden in the combination therapy, compared to meropenem alone, in either the lung or BAL. As such, this sentence warrants clarification.

Line 348-353 – "As expected, local IL-6 and other pro-inflammatory cytokine levels were elevated in BALF at 24 hpi (Fig. 2a; Supplementary Fig. S2), but less so after antibiotic treatment and significantly less after phage treatment, either alone or combined with meropenem." This sentence requires clarification. There are no significant differences in BALF IL-6 between

control and meropenem alone.

Sup Line 54, Fig S2– phages significantly reduced levels of IL23 in BAL, whereas meropenem alone does not – this has not been commented on and may provide insights into the differential responses between treatments. IL-23 plays a role during *Pseudomonas aeruginosa*-induced lung infection in mice.

Fig S2 – are these inflammatory cytokines within the limit of detection of the assay used? Some are produced in very low levels.

In vitro stretch experiments, supernatants used as enumeration of bacterial load – why not lysate? If less bacteria in the supernatant could this be increased internalisation?

Minor:

Figure 1 – despite faster clinical scores, no difference in bacterial burden is evident between meropenem and combined treatment in either the lung or BAL. comment on this.

Line 49 -Some examples of carbapenem resistant pathogens could be included

Line 69 – reword to improve clarity

Line 68- Demonstrate rather than demonstrated

Line 92 - How was phage effectiveness selected?

Line 190 – specifically include details of the gating strategy used for identification of each population.

Line 248 – mechanical lysis was carried out in co-operation with...

Line 350: Re word for clarity and conciseness. - Elevated compared to what?

Line 363 – “As we observed good treatment results” – reword for clarity regarding what results were obtained. Improved clinical scores, rather than reduced bacterial burden.

Figure 4c: 'c' covered by graph

Line 443-444 – phage therapy is more effective. PAO1 were cultured in the absence.

General suggestions regarding discussion.

- Some commentary on target patient population and why this would be of benefit to these patients.

- Some discussion/reference to overuse of other antibiotic classes- Is there potential scope for phages to have similar synergistic effects with other classes.

- Could this be used in other forms of acute infection Community acquired pneumonia?

- Some commentary required regarding the benefits of combination therapy.

Reviewer #4

(Remarks to the Author)

Version 1:

Reviewer comments:

Reviewer #1

(Remarks to the Author)

I appreciate the authors' efforts in revising the manuscript. The point-by-point responses are clear and adequately address the concerns I previously raised. At this stage, I have only minor suggestions for further improvement.

Figure 1: Did the authors compute the clinical score at extubation (0 hpi)? If so, these results should be included in panels (b), (c), and (d).

Figure 5 / Supplementary Figure S5: Please include PFU/ml data to assess the effect of antibiotics on phage replication.

These minor additions will enhance the clarity and completeness of the manuscript.
Thank you for your thoughtful revisions. I look forward to the final version of the manuscript.

Reviewer #3

(Remarks to the Author)

I am satisfied that all concerns raised during peer review have been addressed in the revisions, which has greatly improved the manuscript.

REVIEWER COMMENTS – POINT BY POINT RESPONSE

Response: We sincerely thank the editor and the four reviewers for their time and effort in evaluating our manuscript and providing valuable feedback to enhance its quality and clarity. We deeply appreciate the positive assessment of our work and the thoughtful, constructive suggestions for improvement. In response, we have thoroughly revised the manuscript and addressed all the points raised and added necessary experiments as suggested.

Attached, you will find the point-by-point response, including additional references where necessary. Recommended changes made to the revised manuscript are made in track-change-mode and are detailed in this response letter, highlighted in yellow, respectively.

Reviewer #1 (Remarks to the Author)

Chantal Weissfuss et al. evaluated the combination of a two-phage cocktail with carbapenem antibiotic meropenem for the treatment of ventilator-associated P. aeruginosa pneumonia in mice. They demonstrated that addition of phage therapy improved the clinical recovery of meropenem treated animals, without increasing pro-inflammatory cytokine release. Addition of meropenem on the other hand prevented the emergence of phage-resistant clones. The authors uses unique methods to compare the effects of the various therapies on lung physiology features, such as lung barrier permeability index and function, transepithelial electric resistance, using specific settings, including stretched human primary epithelial cell cultures to mimic ARDS. They observed that adjunctive phage therapy prevents epithelial damage upon P. aeruginosa infection and improved lung epithelial cell integrity when combined to meropenem. The evaluation of phage and antibiotics in experimental models of pneumonia in rodents is not original per se, nor is the evaluation of the host immune response under phage therapy, yet the evaluation of the effect of those therapies on lung epithelia is quite original. However, several major concerns on the methodology of the experiments tempered my enthusiasm.

Response to reviewer#1: We thank the reviewer for the recognition of the originality of our experimental approach. We appreciate the reviewers concerns regarding aspects of our animal model methodology. We believe, the majority of the reviewer's main concerns regarding our methodology may stem from the misunderstanding that the study by Debarbieux *et al.*¹, 2010 is based on the same murine disease model (differences summarized in Table 1). We sincerely apologize that we did not make this clear enough in our initial submission, which has been misleading.

The study by Debarbieux *et al.*¹, 2010 investigated murine *Pseudomonas (P.) aeruginosa* lung infection in male Balb/c mice using a PAK lumi strain, administered via the intranasal route with an infection dose of 1×10^7 CFUs. Our current study investigates a pre-clinical murine model of *P. aeruginosa*-induced ventilator associated pneumonia (VAP). Female C57BL/6 mice were first mechanically ventilated at a high tidal volume (HVt), which induces changes to the alveolar microenvironment enhancing the susceptibility to *P. aeruginosa*. This mimics the clinical scenario in which nosocomial *P. aeruginosa* infections occur in ventilated patients with a compromised epithelial barrier. The ventilated, more susceptible mice were then infected intratracheally with strain PAO1 at a lower dose of 5×10^4 CFUs.

We have carefully addressed the reviewers concerns through additional clarifications, sharing data from our laboratory's establishment of the models and referenced a preprint from our laboratory with a detailed model description. These modifications have strengthened clarity of our manuscript and, we believe fully addressed the issues raised.

Table 1. Summarized main differences between our 2010 and 2024 studies.

Study	Debarbieux et al.¹, 2010	Weissfuss et al., 2024
Animal model	P. ae lung infection	P. ae -induced VAP
Mechanical ventilation	none	HVt (34 mL/kg body weight) 4 hours before infection
Mouse strain and sex	Balb/c, males	C57BL/6, females
P. ae strain	PAK lumi	PAO1
Infection route	Intranasal	Intratracheal
Infection dose	1 × 10 ⁷ CFU/50 µL	5 × 10 ⁴ CFU/20 µL
Treatment	Phage PAK-P1, curative, 2 hpi Phage PAK-P1, preventive, 24 hours before infection	Phage cocktail; Meropenem; combination (4 hpi, 16 hpi)
Monitoring time	Every 24 hours	Every 12 hours
Clinical disease score	none	yes
Analysis time point	Survival, 72 hpi - 16 d Cytokines in BALF, 24 - 48 hpi	- Cytokines, Immune cells in BALF and others, 24 hpi
IL-6, TNFα measurement in BALF	ELISA, DuoSet enzyme- linked immunosorbent assay kits (R&D Systems)	Mouse Inflammation Panel (13-plex with V-bottom plates; BioLegend, San Di- ego, CA, USA)

VAP, ventilator-associated pneumonia

P. ae, *P. aeruginosa*

hpi, hours post-infection

Here are the most important ones.

EXPERIMENTAL PNEUMONIA MODEL

Reviewer#1 comment (C) 1 in a 2010 paper from the same authors, the inoculum chosen was much higher (1x10E7) which resulted in much higher bacterial loads in BAL (1.6x10E8 versus ca. 10E4, Fig. 1f). A dose-finding study justifying for the inoculum choice is thus missing.

Response to Reviewer#1 C1: We understand the reviewer's concern regarding the infection dose being lower than in our earlier study by Debarbieux *et al.*¹, 2010. However, this difference can be explained by several key factors: in our current study, we use a different disease model, namely VAP, which changes the alveolar microenvironment with greater lung damage pre-infection, a different bacterial strain with distinct virulence characteristics, and a different infection route, which results in better bacterial dissemination and disease severity. These factors as well as the use of female C57BL/6 instead of male Balb/c mice, necessitated the adjustment of the infection dose to ensure a reproducible disease model. In the following paragraphs, we provide more details on the model for your convenience and reference our pre-print on the model more prominently in the method section.

Disease model: Unlike previous studies on common murine *P. aeruginosa* pneumonia, our work specifically investigates *Pseudomonas*-induced ventilator-associated pneumonia (VAP). VAP is one of the most prevalent nosocomial infections in intensive care units and has high translational relevance for reducing mortality in critically ill patients.

The purpose of employing this novel murine VAP model, established by Felten *et al.*², is to mimic the infection trajectory as observed for patients with VAP, where pneumonia develops following mechanical ventilation (see Figure 1, Felten *et al.*², 2024 BioRxiv or below for your convenience).

Figure 1: Modified from Felten *et al.*², 2024 Figure 1. Experimental plan of the establishment of *Pseudomonas*-induced VAP in mice.

Previous studies have attempted to develop experimental murine VAP-like models to investigate VAP pathogenesis³⁻⁵. However, unlike the pathophysiological sequence in humans, these studies induced pneumonia by bacterial inoculation in healthy animals before the onset of mechanical ventilation.

Similar to the study by our French collaborators mentioned by the reviewer, significantly higher infection doses were required to induce pneumonia in previously healthy animals compared to those that had undergone mechanical ventilation³⁻⁵. This highlights the impact of pre-existing microenvironmental changes and lung injury by ventilation on infection susceptibility and the need for a model that better reflects the clinical scenario.

Route of infection: While Debarbieux *et al.*¹ performed an intranasal infection, we administered the bacterial inoculum via the tracheal tube at the end of mechanical ventilation to closely mimic the clinical pathology of VAP. In intranasal infection models, bacteria may be partially exhaled and must overcome orotracheal defense mechanisms before reaching the lungs. In contrast, intratracheal infection ensures direct bacterial delivery to the lungs, leading to a more reliable induction of pneumonia and better reflecting the pathogenesis of VAP in critically ill patients.

Bacterial strain differences: In the referred 2010 study they used a different *Pseudomonas* strain PAK lumi, while our VAP model employs PAO1. Given the significant virulence differences between bacterial strains, infection doses required for pneumonia development and effective phage treatment may vary.

In summary, these factors likely explain the differences in infection doses compared to previous studies. During the establishment of our VAP model (Felten *et al.*², preprint in BioRxiv and currently under submission) dose finding experiments were conducted based on prior intranasal infection studies performed in our lab by Opitz *et al.*⁶. Three infection doses (5×10^5 , 5×10^4 and 5×10^3 CFUs) were tested. While the high dose led to the decease of all mice within less than 24 h and the low dose to clearance of the bacteria, but without clinical signs, the intermediate model was suitable to establish an acute pneumonia with clinically relevant signs within 24-hour post-infection (hpi), making it the optimal choice for our study.

We appreciate the opportunity to clarify these points and have revised the manuscript accordingly.

Methods, L 173-175: The establishment of the murine *Pseudomonas*-induced VAP model was previously conducted and detailed in our previous work⁵², including procedure control groups to assess the effects of the ventilation versus infection.

Reviewer#1 C2) a PK study of phages in BAL after i.p. injection is missing (phage without infection group). This group is essential to compute the expected MOIs, to prove that phages underwent a virulent cycle and not that the decrease in bacterial load observed is due to lysis from without.

Response to Reviewer#1 C2: We agree that demonstrating the presence of viable phages after i. p. administration is essential to relate examined results to the designated phage therapy. We apologize for not clearly stating that this aspect was addressed in our previous study (Weissfuss *et al.*⁷, 2023); which specifically examined the pharmacokinetics and immune response to phage treatment using the same phages as those used in our current study. In this study, we demonstrated that phages administered intraperitoneally can disseminate throughout the body and reach the lungs, maintaining stability several days.

To clarify this point, we have now explicitly referenced our previous work more prominently in the methods section.

Methods, L 130-131: In a prior study³¹, we demonstrated that intraperitoneal administration of the phages allowed them to reach the lungs, remain stable for several days, and remain viable.

Reviewer#1 C3) A control group - anesthesia/ventilation only - is missing to assess what is due to the procedure and what is due to the infection in the clinical outcome.

Response to Reviewer#1 C3: Thank you for raising this important point. We have carefully considered the inclusion of additional control groups to ensure scientific rigor while also adhering to regulatory and ethical guidelines, particularly the 3R principles, which emphasize the reduction of animal use.

As mentioned in response to reviewer#1 C1, in our previous study (Felten *et al.*², 2024 BioRxiv), we established the murine VAP model employed in this study. Felten *et al.* demonstrated that infection with *P. aeruginosa* led to distinct pathological changes in mice subjected to high-tidal volume ventilation (HVt), including increased alveolo-capillary permeability, elevated lung and blood leukocyte counts, and higher pulmonary and systemic bacterial loads. These outcomes were compared to procedure control groups, such as mice receiving low-tidal volume ventilation (LVt) or non-ventilated control mice. Thus, this study thoroughly examined the effects of the procedure versus infection on clinical outcomes.

In our current study, in accordance with the 3R principles and following the legally binding guidance from the local animal experiment authorization agency (LaGeSo), we did not replicate these control groups. Furthermore, from a clinical perspective, VAP patients receiving bacteriophage treatment would have undergone similar surgical and ventilation procedures. Therefore, our study specifically aimed to assess the effectiveness of phage therapy in the context of both mechanical ventilation and infection.

We now refer to our earlier study more prominently in the method section to allow the reader to easily find this relevant information.

Methods, L 173-175: The establishment of the murine *Pseudomonas*-induced VAP model was previously conducted and detailed in our previous work⁵², including procedure control groups to assess the effects of the procedure versus infection.

Reviewer#1 C4) Why were the animals killed after 24h? Why was survival not evaluated as secondary outcome? For instance in the previous 2010 study, the animals were followed over 72h. This is absolutely needed to confirm the relevance of the observations.

Response to Reviewer#1 C4: Thank you for your questions. The choice of the 24-hour endpoint was guided by its clinical relevance; it reflects early stages of VAP progression when therapeutic interventions are most impactful in humans. This ensures that our findings are translatable and relevant for guiding treatment strategies.

While we agree that survival experiments could complement our findings, we respectfully disagree with their necessity for confirming the relevance of our observations. Our study focused on evaluating therapeutic efficacy through well-established surrogate markers such as bacterial clearance, inflammatory response modulation, and lung barrier function—all of which are predictive of survival outcomes in this model. This is also in accordance with current clinical observations^{8,9}.

Unlike the 2010 study mentioned by the reviewer, our research employs a distinct murine model adapted to reflect *Pseudomonas*-induced VAP. Disease progression in this model is rapid due to the ventilation procedure pre-infection, resulting in increased susceptibility towards bacterial loads and toxin levels, leading to significant clinical deterioration by 24 hpi. Extending the observation beyond this time point would not provide additional mechanistic insights.

Reviewer#1 C5) *The meropenem dose (1g/kg/d) is clinically irrelevant - for a human being max recommended schemes are around 40 mg/kg/tid = 120 mg/kg/d (or normally 3x1g, max 3x2g for an adult of ca.70-80kg). Authors did not mention the MIC of PAO1 to meropenem either.*

Response to Reviewer#1 C5: Thank you for your thoughtful comment. The decision to use a high-dose meropenem treatment in our murine VAP model was made intentionally based on literature detailing murine *Pseudomonas* infections. Specifically, studies have demonstrated that a treatment of 4 × 500 mg/kg/d (2000 mg/kg/d) in a murine pneumonia model correlates with a high-dose regime of 6 g/d in humans, as shown through PK/PD correlation analyses¹⁰. Since the protein binding of meropenem is five times higher in mice (20%) than in humans (2%)¹⁰, this higher dose ensures sustained antibiotic availability in the murine model. The rationale for the chosen doses is now addressed in the methods and limitation section of the discussion.

In our treatment schedule, we used 10 mg/20 g mouse administered twice at 4 hpi and 16 hpi, with an endpoint at 24 hpi (1000 mg/kg/d), to mimic the clinical scenario of patients receiving 3 × 2 g/d q8h high-dose meropenem. Despite these high antibiotic concentrations, the infection in meropenem-treated VAP mice was not as effectively suppressed as with phage or combination therapy, highlighting the superior efficacy of adjunctive phage therapy.

The MIC of meropenem is 1 µg/mL¹¹⁻¹³, and we have updated this information in the manuscript along with revisions to the graph description, corresponding figure legends, and the main text. Adjustments were also made in the Supplement.

Methods, L186-190: The antibiotic solutions were prepared from intravenous formulations used for humans, reconstituted with sterile NaCl (0,9%; B. Braun, Melsungen, Germany) to a final concentration of 10 mg/mouse per injection (approx. 1 g/kg body weight/day **based on published PK/PD correlation analyses⁵³**). Sterile NaCl (0,9%; B. Braun, Melsungen, Germany) served as control group.

Methods, **(minimum inhibitory concentration; MIC = 1 µg/mL⁵⁴⁻⁵⁶)** L 254-255, L 292, L 310
Results, **(x MIC)** change of µg/mL to corresponding MIC in the whole section

Figure legends, (MIC = 1 µg/mL) Figure 4 L438, Figure 5 L 481, Figure 6 L 536, Figure 8 L 614

Discussion, L 779-782: Notably, since meropenem protein binding is five times higher in mice (20%) than in humans (2%)⁵³, the treatment doses required adjustment to reflect human dose regimens and ensure sustained antibiotic availability in the murine model.

Reviewer#1 C6) *The way the inocula were prepared might not allow for correct assessment of P. aeruginosa virulence. Authors grew isolates up to early exponential phases a time of growth when not all Quorum sensing systems are active (mid exponential phases would have been better). As sign for it, the level of inflammatory cytokines measured are lower compared to the 2010 study (IL-6). As host response is one of the read-out the authors assessed, it is essential to look for a virulent bacterial infection, if not the results on the decreased inflammation induced by phages might well be irrelevant.*

Response to Reviewer#1 C6: Thank you for your detailed comment. We have addressed your points below and hope these clarifications resolve your concerns while providing further insight into our experimental design and approach.

In our study, we successfully induced VAP using the well-characterized laboratory strain *P. aeruginosa* PAO1¹⁴, which is commonly used in murine infection models. The bacterial loads and clinical signs of infection confirmed the establishment of *Pseudomonas*-induced VAP and virulence of the infection. Additionally, IL-6 levels measured in the bronchoalveolar lavage fluid (BALF) of untreated control mice were approximately twice as high as those in treated animals, further supporting the notion of a virulent infection.

We would like to acknowledge that comparing the numerical IL-6 levels between our and the Debarbieux *et al.*¹ study is not a valid way to draw conclusions on divergent virulence between the pathogens causing lung infection. The studies differ in relevant experimental conditions (see Table 1 and Response to reviewer#1 C1). Specifically, the assays used (R&D ELISA versus BioLegend multiplex) and the mouse strains employed (Balb/c versus C57BL/6) differ and impact numerical IL-6 levels. Balb/c mice reached ~ 5 ng/ml IL-6 levels in BALF measured by R&D ELISA (Debarbieux *et al.*¹, 2010) and C57BL/6 mice reached ~ 2 ng/ml IL-6 levels measured by BioLegend multiplex assay (this study by Weissfuss *et al.*, 2024). These differences are to be expected given the different experimental set-ups, and do not warrant conclusions about divergent virulence.

Regarding the preparation of the bacterial inoculum, we intentionally used early exponential-phase bacteria for infection. While we understand that mid-exponential-phase bacteria are often considered more virulent due to quorum sensing activation, the use of early exponential-phase bacteria in our study was based on previous findings showing that *P. aeruginosa* PAO1 can effectively cause pneumonia at both growth phases without significant differences in infection progression or bacterial burden. For example, Kuang *et al.*¹⁵ demonstrated that exponential-phase *P. aeruginosa* can infect mouse lungs similarly to stationary-phase bacteria, showing no major differences in bacterial load or infection outcomes. Furthermore, clinical isolates of *P. aeruginosa* exhibit variability in quorum sensing genes and transcription patterns during different growth phases¹⁶.

For clarity, we now include the corresponding information and reference in the methods section.

Methods, L 115-116: Notably, the PAO1 strain induces pneumonia independent of the inoculum's harvest growth phase^{46,47}.

Reviewer#1 C7) The differences in clinical scores between meropenem and phages observed at 16h is slowly vanishing at 24h, which mandates a prolonged observation time, as beta-lactams need some time to be effective (see below comment on time kill curves).

Response to reviewer#1 C7: Thank you for this remark. You are correct that the clinical scores between the meropenem-only and phage-only groups are no longer significantly different at 24 hpi; however, a strong trend remains. Importantly, there is still a significant difference between the meropenem-only group and the adjunctive phage treatment group, which is the focus of this study.

In vivo, we selected the well-established 24 hpi, which is optimal for this murine infection model and has consistently demonstrated significant differences in clinical scores, bacterial burden, inflammatory markers, and lung barrier function. Moreover, our *in vitro* studies showed that even extending the treatment duration did not improve bacterial clearance with meropenem monotherapy (see Fig. S5). Overall, our findings highlight that adjunctive phage therapy offers the best therapeutic outcome in our experimental model compared to both monotherapies and the control group.

In response to your comment Reviewer#1 C9 – time kill curves Fig. 5 (find details below in response to comment), we performed additional *in vitro* experiments, which showed that prolonged incubation times of adjunctive phage treatment remained most efficient (see Results, L 505-516 and Fig. S5).

However, we agree that restriction to one time point can be considered as a limitation and highlight this more clearly in the limitation section of the discussion.

Results, L 505-508: Even with an extended incubation time of 24 h, adjunctive phage treatment demonstrated the highest efficacy compared to monotherapies *in vitro* (Supplemental Fig. S5). In the high-dose combination treatment ++++ (Meropenem 5 × MIC, Phages MOI 0.5) the bacterial load remained below the detection limit.

Discussion, L 769-773: Additionally, assessing additional time points could offer a more comprehensive understanding of both innate and adaptive immune responses while enabling the tracking evaluating earlier or later time points could offer a more detailed analysis of innate and adaptive immune responses and track the emergence of resistant bacterial clones emerging over time *in vivo*.

Reviewer#1 C8) PFU assessment in tissue samples in parallel to CFU is missing. My question would be the following: does meropenem reduce the phage replication by decreasing the number of hosts available?

Response to reviewer#1 C8: It seems there may be a misunderstanding; please excuse any confusion. We indeed conducted the PFU analysis of tissue samples alongside the CFU analysis. Please refer to Fig. 1e.

To enhance clarity and understanding, we have added additional headings to the sub-figures: Fig. 1e, "Phage Load"; Fig. 1f, "Bacterial Load (Local)"; and Fig. 1g, "Bacterial Load (Systemic)."

With regard to the question raised by the reviewer: We did not observe reduced phage replication upon meropenem treatment. Instead, our results suggest that meropenem limited the outgrowth of newly emerging, likely phage-resistant bacterial clones. Phage loads showed no differences between the phage-only and combination group across all sampled organs, regardless of bacterial load. Furthermore, phage titers across all sampled organs remained similar and did not show significant changes, irrespective of whether mono- or combination treatment was applied (see Fig.1 e).

TIME-KILL CURVES (Fig.5)

Reviewer#1 C9) *I questioned the rationale of the time-line chosen for this experiment, which cannot assess meropenem efficacy. Indeed, time over MIC is the PK/PD index to be targeted for beta-lactam efficacy. Ending the experiment after 3 hours is not enough for meropenem to be efficient, as demonstrated by the only minor reduction of the bacterial load. It is well known that bactericidal effect of phages is fast, but the way the results are presented over such short period of time might be misleading for the comparison with meropenem.*

Response to reviewer#1 C9: Thank you for this valuable remark. We agree that a 3-hours' time course is not optimal to fully assess meropenem's efficacy, given that beta-lactams rely on time above MIC. To address this, we followed your advice and extended our infection assays by 8-times to 24 hours (see Supplementary Figure S5).

The aim of our study was to evaluate if combination therapy of meropenem and phages offers advantages over phage-only or meropenem-only treatments. Specifically, we observed: even with high-dose meropenem (50 × MIC) treatment, there was a slower decrease in bacterial load, which failed to fully clear the bacteria as effectively as the high-dose combination therapy (Meropenem 5 × MIC, Phages MOI 0.5). In the mid-dose combination (Meropenem 0.5 × MIC, Phages MOI 0.05), a regrowth of likely phage-resistant bacteria was observed, but only after 9 hpi. At this time point, bacterial concentrations were still lower than those seen with mono-antibiotic treatment, even though the antibiotic concentration was 100 times lower.

The new data was included in Fig. S5 and corresponding figure legend and results updated accordingly.

Results, L 497-500: The adjunctive phage therapy was most effective, including fast bacterial killing and prevention of regrowth of **likely** phage-resistant clones **for up to 3 h in this experimental setting** (Fig. 5b), matching results in the presence of epithelial cells (see Fig. 4b).

Results, L 505-516: **Even with an extended incubation time of 24 h, adjunctive phage treatment demonstrated the highest efficacy compared to monotherapies *in vitro* (Supplemental Fig. S5). In the high-dose combination treatment ++++ (Meropenem 5 × MIC, Phages MOI 0.5) the bacterial load remained below the detection limit. In contrast, a slight increase in likely phage-resistant bacteria was observed with the mid-dose combination ++ (Meropenem 0.5 × MIC, Phages MOI 0.05), but only after 9 h. However, the high-dose meropenem monotherapy (50 × MIC) failed to significantly reduce the bacterial load as effectively as the adjunctive therapy, even at later time points. At the 9 h-time point, bacterial concentration remained higher than under both combination therapies and only decreased to approximately 10³ CFU/mL by 24 hpi. As expected, bacterial load still increased in a time-dependent manner under phage monotherapy (MOI 5). These findings highlight the superior efficacy of adjunctive phage treatment, enabling the use of reduced concentrations of both meropenem and the phage cocktail.**

Figure S5: Long term analysis and direct comparison of the bacterial clearance in the Infection assay with different antibacterials. For the infection assay PAO1 were cultured in the absence (control) or presence of meropenem (MIC = 1 µg/mL), the Pseudomonas-specific phage cocktail, or the combination of both in different concentrations as indicated. **(a) Colony-forming units (CFU/mL) according to time.** Data are expressed as means of 2-3 independent experiments ± SD. Dotted line reflects detection limit. **(b) Combined results** are shown as means ± SD of bacterial load at different time points, as determined by two-way ANOVA with Dunett's multiple comparisons test: *p < 0,05, **p < 0,01, ***p < 0,001, ****p < 0,0001; n = 3. Dotted line reflects detection limit. MIC, minimum inhibitory concentration; MOI, multiplicity of infection

Reviewer#1 C10) *I would use multiple of MIC rather than meropenem concentrations in the legend.*

Response to reviewer#1 C10: Thank you for this helpful remark. We have added the MIC value for meropenem (1 µg/mL) to the manuscript and made the necessary adjustments to improve the clarity in the graph description. Please refer to Response to reviewer#1 C5) above.

Reviewer#1 C11) *meropenem concentrations as high as 50 mg/L are not clinically relevant. Usual through target levels are supposed to be around 5-10 mg/L. one might observe peak around 50 mg/L but as continuous/prolonged infusion is being currently recommended, such concentrations are becoming less common.*

Response to reviewer#1 C11: Indeed, we acknowledge that the concentration of 50 µg/mL meropenem was used as a high-dose (positive) control, exceeding typical clinical concentrations. This decision was made intentional to serve as a robust control in our *in vitro* studies. Additionally, we administered only a single dose in these experiments without employing a continuous infusion. Given the static nature of the medium, the concentration of the antibiotic is expected to remain relatively stable throughout the experiment, thereby extending its activity and relevance under these experimental conditions.

Reviewer#1 C12) *An observation time beyond 3 hours might have highlighted the emergence of phage resistant mutants. When this phenomenon was observed, were those mutants analyzed (Fig4b)?*

Response to reviewer#1 C12: Thank you for your question. No, we did not analyze potential phage-resistant mutants further in our study. However, this is a very interesting and exciting direction that we would like to address in more detail in our future research. Based on our observations of bacterial regrowth under phage treatment, we hypothesize that the emerging bacterial clones may partially or even fully phage-resistant. This hypothesis aligns with findings from other published studies^{17,18} and a related study conducted by our co-authors, which verified such outcomes¹⁹.

In addition to previously mentioning the need for further exploration of likely bacterial mutants that emerged in our study (see Discussion, L714-717), we have also added a paragraph in the limitations section.

Discussion, L 773-775: **Further investigation is needed to validate our findings and determine whether bacterial mutants arise under adjunctive phage therapy or not, which would underscore the value of phage treatment as an adjuvant to conventional antibiotics.**

Reviewer#1 summary. *In summary, while authors looked at lung cell injury using advanced assays, there are several issues in the experimental design, such as short observation time, missing control groups (e.g. phage only, procedure only), Dose-finding/PK studies, PFU assessment for animal studies, which questioned the relevance of the results. In addition, it is not clear which therapy is adjunctive to which one, which makes the reading sometimes confusing (e.g. line 426-428 the adjunctive phage therapy is not preventing the outgrowth of phage-resistant clones, but rather meropenem). Authors should make clear, what they considered standard of care, what they consider as adjunctive therapy, etc.*

Response to reviewer#1 summary: Thank you for your clear summary of your main concern. We greatly appreciate the opportunity to clarify these points. Upon reviewing our response letter, you will see that we have addressed, and we believe resolved, all of the issues

raised—many of which stemmed from a misunderstanding of the experimental procedures. We take full responsibility for this and apologize for any inconvenience this may have caused.

To further enhance clarity, we have included more detailed information regarding our treatment groups: the standard-of-care treatment involving monotherapy with meropenem, the monotherapy with the specific phage cocktail, and the adjunctive phage therapy, where phage treatment was applied in combination with meropenem. Our study underscores the significant potential of using phages as an adjunct to classical antibiotic therapy in treating MDR-bacteria induced VAP.

Results, L 331-333: Treatment occurred at 4 and 16 hpi with either meropenem (monotherapy; standard-of-care), the phage cocktail (monotherapy), a combination of both (adjunctive phage therapy), or PBS as control.

Results, L 461-464: Notably, the adjunctive phage therapy combined the fast and specific killing effect observed with specific phages and the prolonged antibacterial potential of the standard-of-care treatment with meropenem, prevents limiting the outgrowth of likely phage-resistant clones.

Discussion, L 631-633: Here, we show in a *Pseudomonas*-induced murine VAP model that combined treatment with combining the standard-of-care antibiotic meropenem and with a *Pseudomonas*-specific phage cocktail [...].

Reviewer #2 (Remarks to the Author)

The manuscript by Weissfuss et al describes a comprehensive study of phage and antibiotic therapy against P. aeruginosa in a VAP model, including an initial in vivo study followed by extensive in vitro studies. This is a useful study that benefits from a tight focus on a phage combination and one antibiotic, which allows for more depth of study. In clinical use, phage are often used alongside standard antibiotics and the interaction of these two agents is important for development of phages as therapeutics. The manuscript is generally well-written and well-presented, with relatively minor revisions needed. One issue that must be addressed is a better description of how the phages were prepared for in vivo and tissue culture experiments, noted below. The discussion section is very comprehensive but rather lengthy and could also benefit from some condensing.

Response to reviewer#2: We would like thank the reviewer #2 for their positive feedback and careful evaluation of our manuscript. As suggested, we condensed the discussion and addressed all comments, which has helped further improving our manuscript.

Reviewer#2 C1-3) L 41: Ventilator-associated pneumonia; 2) L47: in recent years, 3) L 50: selecting for carbapenem-resistant pathogens

Response to reviewer#2 C1-3: Thank you for the suggestions. We have revised the text of the introduction accordingly.

Introduction, L 41: Ventilator-associated pneumonia (VAP) is one of the most common hospital-acquired infections¹.

Introduction, L 44-47: It extends the time on mechanical ventilation and hospital stay, thereby increasing both morbidity and healthcare costs, and is associated with high mortality rates ranging from 13%² to 42%³ among ICU patients or even 47%⁴ among COVID-19 patients in the last recent years.

Introduction, L 48-52: Although carbapenems, e. g. meropenem, are recommended as a first-line treatment for VAP when antibiotic resistance is suspected⁵, their use carries the risk of inducing selecting for carbapenem-resistant pathogens, including *P. aeruginosa*, *Acinetobacter baumannii*, *Klebsiella pneumoniae* and *Enterobacter* species^{6,7}.

Reviewer#2 C4) L 117-124: how were the phages prepared for in vivo and tissue culture use, were they purified to remove cell debris, media and bacterial endotoxin? Unpurified phage lysates usually contain large amounts of endotoxin (thousands of EU/ml) that can cause inflammatory responses depending on how they are administered.

Response to reviewer#1 C4: We apologize for any concerns. All our *in vivo* and *in vitro* experiments were conducted using highly purified phage preparations to prevent any inflammatory responses caused by endotoxins, cell debris, or other contaminants. We have included the necessary information in the methods section more detailed and hope this clarification addresses any uncertainties.

Methods, L 127-130: Additionally, the individual phage suspensions were highly purified through chromatography, and endotoxins were removed by the supplying institutions to low and non-immunogenic levels, as previously reported³¹.

Reviewer#2 C5) L 334-336: this is not what is shown in the figure, there is no statistical significance in the differences between groups.

Response to reviewer#2 C5: Thank you for this remark, and we apologize for any confusion. In our murine VAP model, we observed a significant improvement in body temperature at 16 hpi with phage and combination treatments, which remained elevated as a trend at 24 hpi (Fig. 1c). No significant differences were observed in body weight changes over time across the different treatment options (Fig. 1d). However, as the clinical disease score incorporates additional clinical signs beyond body weight loss and body temperature (see Supplementary Table S1, Fig. 1b), we used it as an important reference for therapeutic outcomes and recovery from VAP. To enhance clarity, we have revised the text accordingly.

Results, L 354-358: In line with this, a more marked recovery of body temperature was observed in the two groups receiving phages, with significant improvement at 16 hpi and a continued upward trend at 24 hpi in the two groups receiving phages was observed (Fig. 1c). This indicates a faster recovery clinical improvement from VAP compared to the control and meropenem only groups.

Reviewer#2 C6) L 454, 455: use “to” instead of “up to” here.

Response to reviewer#2 C6: We have revised the text accordingly.

Results, L 489-493: Within the first 30 min of incubation, the highest concentration of phage cocktail (MOI 5) could significantly reduce the bacterial load up to approx. 10^2 CFU/mL, whereas the highest concentration of meropenem (50 x MIC μ g/mL) could reduce bacterial load up to approx. 10^6 CFU/mL, a minor reduction compared to the untreated control (approx. 10^8 CFU/mL) (Fig. 5b, Supplemental Fig. S5b).

Reviewer#2 C7) L 457-459: is it known if the two phages use the same receptor, or if they exhibit cross-resistance (i.e., bacterial mutation to resistance against one phage also makes them resistant to the other phage)?

Response to reviewer#2 C7: Thank you for the insightful question; we agree that this is an interesting point. The characterization of the phage cocktail phages was performed by our collaborators and co-authors at ITEM Fraunhofer (Braunschweig, Germany) (manuscript in

preparation). In addition, another study by our colleagues and co-authors specifically focused on evaluating interplay of the cocktail's phages *in vitro*¹⁹. These studies showed a) that PAO1 exhibited cross-resistance towards both phages, b) that both phages interact with LPS^{20,21} and c) that observed phage competition was based on differing latency times *in vitro*.

Notably, despite this, the phages remained highly effective in our pre-clinical *in vivo* VAP model.

To improve clarity, we revised the text accordingly.

Results, L 495-497: **In this study, the efficacy of the phage cocktail was confirmed, regardless of the individual contributions of single phages, despite the likelihood of receptor competition or cross-resistance.**

Reviewer#2 C8-9) L 497: exceeding, 9) L 502: In contrast,

Response to reviewer# C8-9: Thank you for the suggestions. We have revised the text accordingly.

Results, L 548-550: The normalized resistance decreased over time up to approx. 0.8 after 12 h, ~~excelling~~ **exceeding** epithelial cell layer disruption provoked by high dose (5 µg/mL) LPS control (Fig. 6b).

Results, L 554-555: ~~Contrary~~ **In contrast**, the supernatants of the phage-treated PAO1-LC (MOI 5) had no impact on the integrity of the cell layer regardless of the sampling time point (Fig. 6d and f).

Reviewer#2 C10) Fig. 1: clinical score includes temperature and weight, so it seems like these factors are being counted twice here.

Response to reviewer#2 C10: Indeed, the clinical score incorporates body temperature and body weight changes, along with other clinical features of murine pneumonia. For further details, please refer to Supplementary Table S1.

The clinical score was developed in collaboration with local governmental authorities and animal care officers to accurately assess disease burden and identify animals reaching humane endpoint criteria for euthanasia. It was designed to be as objective as possible, incorporating key disease parameters such as breathing patterns, grooming, and social behavior, alongside objective numerical values like body temperature and body weight change. Based on our experience, this scoring system effectively reflects disease severity and is suitable for representing "disease" in graphical data. Additionally, we provide numerical body temperature and body weight values to facilitate cross-study comparisons, as not all research groups use advanced disease scoring systems. Since part of the score relies on observational monitoring, variations may exist between labs and regulatory guidelines. We recognize the importance of this information and have clarified it further in the manuscript.

Methods, L 166-168: **Body temperature and body weight are represented in the clinical disease score and individually (Fig. 1) to facilitate cross-study comparisons.**

Results, L 343-344: Graphs displaying **(b)** murine clinical disease score (see Supplementary Table S1), **including (c)** body temperature and **(d)** body weight change at indicated time points.

Reviewer#2 C11) Fig. 4: indicate what infection control and medium control mean here

Response to reviewer#2 C11: Thank you for the suggestion. We have revised the figure legend accordingly.

Figure legend Fig. 4, L 447-448: Cells without any infection or treatment serves as medium control, infected cells (MOI 1) without any treatment serves as infection control.

Reviewer #3 (Remarks to the Author):

“Adjunctive phage therapy improves antibiotic treatment of Ventilator-Associated-Pneumonia with Pseudomonas aeruginosa” (NCOMMS-24-52003) by Weissfuss et. al.

Comments for Author

Weissfuss et al. aimed to investigate the effectiveness of combining antibiotics, specifically, meropenem with phages to improve their effectiveness in a model of ventilator-associated pneumonia (VAP), induced by the Gram-negative bacterium Pseudomonas aeruginosa.

The authors convincingly demonstrate the enhanced effectiveness of antibiotic-phage combination therapy over that of traditional antibiotic monotherapy for the treatment of VAP. Specifically, in a VAP mouse model, in initial experiments they demonstrate that mice receiving combination therapy have a faster clinical recovery, compared to those of monotherapy, albeit no differences in lung bacterial burden are evident. They further show that local inflammatory mediator (IL-6), as expected, increased in the VAP model, but is significantly reduced following phage therapy alone, or in combination with meropenem.

They further demonstrate that lung barrier function and systemic organ injury is improved by phage treatment in VAP.

These in vivo findings were corroborated in vitro, whereby the authors demonstrate a dose dependent reduction in bacteria using combination therapy, which displayed the most significant bacterial killing capacity at 5ug/ml meropenem; lower than the usual minimum effective dose of 50ug/ml. These studies were also complemented by in vitro human data where it is demonstrated that combination therapy prevents bacterial-induced damage to human primary epithelial cells.

Taken together, the study by Weissfuss et. al. clearly demonstrates that adjunctive phage therapy improves antibiotic treatment during P. aeruginosa-induced VAP. The study and methodological approaches appear to be technically sound and in sufficient detail to be reproduced. The manuscript includes high-quality data that will be of interest to microbiologists, those studying host-pathogen interactions, and antimicrobial resistance, in addition to related fields including airway innate immunity and antibiotic drug delivery approaches. The authors present data to support an alternative therapeutic approach to acute VAP that has improved efficacy and has fewer adverse effects. Given the increased acquisition of multidrug resistance strains, there is an urgent and unmet need for novel treatment options for critically ill patients' development VAP. Accordingly, this research is important and timely. Despite this, the mechanism behind the synergistic effects of combination therapy presented herein is currently unknown. The use of clinical strains, rather than the laboratory strain PAO1 would have significantly improved the manuscript. For example, performing selected key experiments, similar to those in Fig 1 and 2 with a clinical isolate to demonstrate clinical relevance/effectiveness, would have been greatly beneficial. However, the authors do acknowledge this as a limitation to the study and the main aim to assess if adjunctive therapy could improve antibiotic treatment during VAP has been achieved.

Response to reviewer#3: We greatly appreciate the positive feedback on our manuscript and would like to thank reviewer #3 for the considerable effort invested in its review. The valuable insights provided have allowed us to significantly improve our work and we hope that it is now well suited for publication.

Reviewer#3 Major:

Reviewer#3 C1) Meropenem alone and combination reduce bacterial burden to a similar level (no difference between the two groups), yet combination said to be more effective, what defines recovery?

Response to reviewer#3 C1: Thank you for this remark. In our murine VAP model, we defined “recovery” from *Pseudomonas*-induced VAP based on any amelioration of read-outs that indicate disease severity, this includes bacterial burden, but also inflammatory markers and clinical signs of disease. The clinical signs include changes in body temperature and body weight over the treatment period, as well as general behavioral observations such as breathing patterns and fur appearance (detailed in Supplementary Table S1). We realize that the term “recovery” is strongly associated with a full disease cure. To enhance clarity and better reflect the scope and limitations of our animal model, we have adjusted our wording throughout the manuscript to “clinical improvement” or “disease amelioration”, respectively.

Nonetheless, while the reviewer is correct that meropenem monotherapy reduced bacterial levels to the same extent as adjunctive phage therapy in our murine VAP model, it failed to prevent the increase in inflammatory mediators or lung barrier damage. In contrast, combination therapy demonstrated synergy: phages effectively limited host inflammation and rapidly killed bacteria, while meropenem provided sustained bacterial elimination. Given that controlling host inflammation is crucial for preventing severe disease progression in pneumonia patients, as recently published by Dequin *et al.*²², our results suggest that this synergistic mechanism may ultimately lead to better outcomes in VAP. By promoting rapid bacterial clearance, reducing inflammation, and minimizing lung barrier damage *in vivo*, this approach highlights the superior effectiveness of combination therapy.

We include this relevant aspect now more prominently in the discussion.

Discussion, L 667-672: **Importantly, [---] the regulation of host inflammation is crucial for preventing severe disease progression in pneumonia patients⁶³. In our study, adjunctive phage treatment resulted in a significant reduction of pro-inflammatory cytokines (IL-6, IL-23, TNF α , IL-1 α) in BALF, suggesting that this synergistic mechanism may ultimately lead to better outcomes in patients with VAP.**

Reviewer#3 C2) Figure 2 – The inclusion of BALF cytokines, aside from IL-6 would be beneficial, particularly those which change in the VAP model and specifically with phage therapy – IL23

Response to reviewer#3 C2: Thank you for this suggestion, which we fully agreed with. To provide a clearer understanding of the beneficial effects of adjunctive phage treatment, we have rearranged Fig. 2 and Supplementary Fig. S2. Additionally, we have revised the manuscript to reflect these changes. Please also refer to the comment below.

Results, L 370-373: As expected, ***Pseudomonas*-induced VAP led to an increase in local IL-6 and other pro-inflammatory cytokine levels, including IL-23, TNF α and IL-1 α were elevated in BALF at 24 hpi (Fig. 2a; Supplementary Fig. S2).**

Figure Legend Fig. 2, L 383-384: (a) Level of pro-inflammatory cytokines IL-6 (pg/mL), IL-23 (pg/mL), TNF α (pg/mL) and IL-1 α (pg/mL) in BALF of mice with VAP mice analyzed at 24 hpi.

Reviewer#3 C3) Line 149 – female mice are used for all experiments, but this is not justified. Male and Female mice have differing levels of susceptibility to respiratory bacterial infection and as such, justification as to why only female mice are used throughout this study is warranted.

Response to reviewer#3 C3: Thank you for this remark, we agree that an additional comparison between female and male mice would be informative. We are aware that the degree of pneumonia severity differs between male and females^{23–25}.

The exclusive use of female animals in this study is based on the fact that the murine model of *Pseudomonas*-induced VAP was initially established at our institution using female C57BL/6 mice. In this model, the clinical presentation of pneumonia and VAP in female C57BL/6 mice is consistently reproducible. Furthermore, our research group has prior experience with the immune response to phage therapy, with earlier studies also conducted in female C57BL/6 mice. Using male test animals would significantly limit comparability with these studies and require an entirely new series of establishment and confirmation experiments. Due to different pneumonia severity degrees, male and female mice cannot be compared directly.

To improve clarity, we have added the justification in the study limitation section.

Discussion, L 767-769: Moreover, the study is restricted to female mice to reduce variability in pneumonia severity introduced by sex⁷⁶⁻⁷⁸. Potential sex-specific effects need to be addressed in future studies.

Reviewer#3 C4) Line 167 – meropenem is usually administrated i.v. Herein, i.p injections are used. Justification as to why this approach is used and how it translates clinically is required.

Response to reviewer#3 C4: Our decision to use intraperitoneal (i. p.) rather than intravenous (i. v.) injections was based on two key considerations. First, Oshima *et al.*¹⁰ demonstrated that i. p. and i. v. administration of meropenem resulted in comparable systemic distribution. Second, in our therapeutic murine VAP model, repeated injections were required, and i. v. administration would have necessitated infrared tail treatments. This process would not only be more time-consuming but could also increase stress in the animals, potentially influencing disease severity or clinical improvement. Therefore, we opted for intraperitoneal antibiotic administration in our study.

To improve clarity, address this point now in the limitation section.

Discussion, L 775-779: Unlike the intravenous (i. v.) administration of meropenem used in clinical settings, we chose intraperitoneal (i. p.) administrations in our study as both routes achieve comparable systemic distribution⁵³ while reducing handling time and experimental stress for the animals, thereby minimizing potential effects on disease severity or clinical improvement.

Reviewer#3 C5) Line 413 – Figure 4. do dot plots represent the mean of independent experiments, or as stated technical duplicates? If so, is this data from one single experiment? N numbers should be clarified.

Response to reviewer#3 C5: Thank you for the remark. To improve the clarity, we revised the figure legend accordingly.

Figure legend Fig. 4, L 444-446: Data are shown as means of experiments with box plots depicting median, quartiles, and range, as determined by two-way ANOVA with Tukey's multiple comparisons test: * $p < 0,05$, ** $p < 0,01$, *** $p < 0,001$, **** $p < 0,0001$; $n = 3$ independent experiments, while each condition was run in duplicates per experiment.

Reviewer#3 C6) Figure 5 b and c – these figures are quite busy, with multiple groups displayed on the graphs. Graphs could be reformatted or enlarged to improve clarity for the reader

Response to reviewer#3 C6: Thank you for the suggestion. We have revised Fig. 5 and enlarged the graphs (b) and (c) to improve clarity.

Reviewer#3 C7) Line 490. Figure 6 – As per Figure 4, it is unclear how many independent experiments were performed. This should be clarified throughout all figures. Data presented as the mean of independent experiments or are these technical duplicates?

Response to reviewer#3 C7: Thank you for the remark. To improve the clarity, we revised the figure legend accordingly.

Figure legend Fig. 6, L 539-542: [...] $n = 3-4$ independent experiments, while each condition was run in triplicates per experiment.

Reviewer#3 C8) Line 363 & Line 345 – “As we observed good results as the primary site of infection” – No significant differences in bacterial burden, one of the main readouts of infection status, was evident in the combined therapy group when compared to individual therapy (meropenem alone). Concurrently, no significant differences in PMN or leukocytes are evident in either treatment groups (meropenem alone, versus combination).

Response to reviewer#3 C8: Thank you for pointing this out. We agree that the bacterial burden was similarly reduced across the different treatment regimens, and the immune cell properties also remained comparable. However, we observed a significant reduction in the clinical disease score over time, which was greater and faster with adjunctive phage therapy. Additionally, inflammatory markers in the BALF and lungs were significantly reduced in phage-treated mice with VAP. These findings were later supported by evidence of reduced lung barrier damage compared to the control and meropenem-only groups, further highlighting the benefits of adjunctive phage therapy.

To enhance clarity, we have revised the text at the recommended lines accordingly.

Results, L 392-395: As we observed good positive treatment results at the primary site of infection in terms of clinical improvement with reduced levels of pulmonary inflammatory markers upon phage monotherapy and adjunctive phage treatment, we next investigated lung barrier damage and systemic organ injury.

Reviewer#3 C9) Line 245 – “mice that received adjunctive phage therapy benefited the most, with improved clinical scores and lowest bacterial burden compared to either meropenem or phage cocktail therapy.” While it is acknowledged that combine therapy resulted in improved clinical scores, there were no significant differences in bacterial burden in the combination therapy, compared to meropenem alone, in either the lung or BAL. As such, this sentence warrants clarification.

Response to reviewer#3 C9: Thank you for this remark. To improve clarity, we revised the text accordingly.

Results, L 367-369: Taken together, mice that received adjunctive phage therapy benefitted the most, with significantly improved clinical scores and lowest bacterial burdens compared to either meropenem or phage cocktail therapy alone.

Reviewer#3 C10) Line 348-353– “As expected, local IL-6 and other pro-inflammatory cytokine levels were elevated in BALF at 24 hpi (Fig. 2a; Supplementary Fig. S2), but less so after antibiotic treatment and significantly less after phage treatment, either alone or combined with meropenem.” This sentence requires clarification. There are no significant differences in BALF IL-6 between control and meropenem alone.

Response to reviewer#3 C10: Thank you for highlighting this point. We have revised Fig.2a and Supplementary Fig. S2 and also revised the text accordingly.

Results, L 370-379: As expected, *Pseudomonas*-induced VAP led to an increase in local IL-6 and other pro-inflammatory cytokine levels, including IL-23, TNF α and IL-1 α were elevated in BALF at 24 hpi (Fig. 2a; Supplementary Fig. S2). , but less so after antibiotic treatment and significantly less after phage treatment, either alone or combined with meropenem. Following phage therapy, either alone or combined with meropenem, cytokine levels were significantly lower compared to the untreated control group. In contrast, monotherapy with meropenem resulted in a trend toward reduced levels of IL-6 and IL-23 and significantly lowered levels of TNF α and IL-1 α compared to the control group. However, these levels remained higher compared to those observed in the phage-only and combination treatment groups.

Reviewer#3 C11) Sup Line 54, Fig S2– phages significantly reduced levels of IL23 in BAL, whereas meropenem alone does not – this has not been commented on and may provide insights into the differential responses between treatments. IL-23 plays a role during *Pseudomonas aeruginosa*-induced lung infection in mice.

Response to reviewer#3 C11: Thank you for this important remark. We have now included IL-23 in the results section (please see Fig. 2 and refer to reviewer#3 C2 and C10 above) and added a paragraph in the discussion to address this relevant point in more detail.

Discussion, L 667-679: Importantly, [---] the regulation of host inflammation is crucial for preventing severe disease progression in pneumonia patients⁶³. In our study, adjunctive phage treatment resulted in a significant reduction of pro-inflammatory cytokines (IL-6, IL-23, TNF α , IL-1 α) in BALF, suggesting that this synergistic mechanism may ultimately lead to better outcomes in patients with VAP. For example, IL-23, which promotes neutrophil recruitment and pro-inflammatory gene expression in pulmonary epithelial cells via IL-17 signaling^{64,65}, was reduced following phage-only and combination treatments, as corroborated by a trend toward reduced IL-17 levels. Although neutrophil counts in BALF and lungs remained similar across all groups, phage-treated mice showed significant protection against lung barrier damage. Collectively, these results suggest that faster bacterial clearance by (adjunctive) phage therapy minimized inflammatory mediator release without affecting neutrophil recruitment.

Reviewer#3 C12) Fig S2 – are these inflammatory cytokines within the limit of detection of the assay used? Some are produced in very low levels.

Response to reviewer#3 C12: Thank you for your question. Yes, some of the cytokines measured using the LEGENDplex™ Mouse Inflammation assay fall within the limit of detection. The detection limit is indicated in each graph by the dotted line, respectively. The information was provided in the legend of Fig. 2, but not Fig. 3, yet. We have adapted this.

Figure legend, Fig. 3, L 414: Dotted lines represent the detection limit.

Reviewer#3 C13) *In vitro* stretch experiments, supernatants used as enumeration of bacterial load – why not lysate? If less bacteria in the supernatant could this be increased internalization?

Response to reviewer#3 C13: Thank you for this notion. Indeed, one of the virulence factors of *P. aeruginosa* is its ability to adhere to and invade host cells, including human lung epithelial cells. We specifically addressed this point in a preliminary study conducted as part of S. Rotter's master thesis²⁶ under the supervision of the corresponding author PD Dr.-Ing. G. Nouailles. Here, we did not observe a significant impact by internalization of the human primary alveolar epithelial cells (HPAEpiC) conducted via bacterial load in supernatants or cell-adherent (see Figure 4, master thesis S. Rotter²⁶, unpublished data). The bacterial load in the supernatants (S) was more than twice as high as that of cell-adherent (C) bacteria, which represent internalized bacteria. Based on these findings, we opted to analyze only the supernatants from our *in vitro* stretch (unphysiological stretch) experiments to assess bacterial load at different time points. Since our primary focus is on evaluating the therapeutic efficacy of adjunctive phage treatment, we attribute the observed decrease in bacterial load to the treatments themselves.

Figure 4. Adapted from S. Rotter's master thesis²⁶. Comparison of bacterial load (*P. aeruginosa* strain ATCC 27853 [CFU mL⁻¹]) found internalized, e. g. cell adherent (C), or in supernatants (S) at 2 hours post infection (hpi) and 4 hpi of HPAEpiCs after 24 h physiological (phys) and unphysiological (unphys) stretch along with static control (stat). conditions. Data are depicted as Mean + SEM of n = 10 (24 h + 2 hpi), n = 8 (24 h + 4 hpi). Statistical analysis was performed through one-way Kruskal Wallis test; ns= non-significant, *p < 0.05, **p < 0.01.

Reviewer# 3 Minor:

Reviewer#3 C14) Figure 1 – despite faster clinical scores, no difference in bacterial burden is evident between meropenem and combined treatment in either the lung or BAL. comment on this.

Response to reviewer#3 C14: Thank you for highlighting this. Indeed, we demonstrated that the bacterial burden was reduced to the same extent with meropenem monotherapy as with adjunctive phage treatment. However, while meropenem effectively reduced bacterial

loads, it did not prevent the increase in inflammatory mediators or protect against lung barrier damage. The clinical score described in Fig. 1 reflects the overall disease burden of the animals which is driven by pathogen burden and inflammatory status. We now include this more clearly in the manuscript.

Results, L 333-335: The clinical status, reflecting disease burden—comprising both pathogen load and inflammation—was monitored at 4, 16 and 24 hpi, and mice were sacrificed at 24 hpi for analysis (Fig. 1a).

Reviewer#3 C15) Line 49 -Some examples of carbopenam resistant pathogens could be included

Response to reviewer#3 C15: We added the examples as suggested.

Introduction, L 48-52: Although carbapenems, e. g. meropenem, are recommended as a first-line treatment for VAP when antibiotic resistance is suspected⁵, their use carries the risk of inducing selecting for carbapenem-resistant pathogens, including *P. aeruginosa*, *Acinetobacter baumannii*, *Klebsiella pneumoniae* and *Enterobacter* species^{6,7}.

Reviewer#3 C16) Line 69 – reword to improve clarity

Response to reviewer#3 C16: We have revised the text accordingly.

Introduction, L 70-72: However, although case reports highlight the effectiveness in individual patients, no controlled clinical trial has yet demonstrated successful therapeutic application. ~~while case reports demonstrated individual effectiveness in single patients, so far no controlled clinical trial has reported successful therapeutic application.~~

Reviewer#3 C17) Line 68- Demonstrate rather than demonstrated

Response to reviewer#3 C17: Thank you, we corrected it. Please refer to Response to reviewer#3 C16.

Reviewer#3 C18) Line 92 - How was phage effectiveness selected?

Response to reviewer#3 C18: The phages were previously selected against various *P. aeruginosa* strains by DSMZ (Braunschweig, Germany) and ITEM Fraunhofer (Braunschweig, Germany) using the Routine Test Dilution method, as previously described²⁷. We have incorporated the necessary information and revised the text accordingly.

Methods, L 125-127: The phages were previously selected against various *Pseudomonas aeruginosa* strains by the providing institutions using the Routine Test Dilution method, as described elsewhere⁴⁸.

Reviewer#3 C19) Line 190 – specifically include details of the gating strategy used for identification of each population.

Response to reviewer#3 C19: Thank you for the remark; however, there might be a misunderstanding. We have already included a reference to Supplementary Fig. S1 for additional details regarding the gating strategy. To enhance clarity, we have rearranged the text accordingly.

Methods, L 210-211: For the detailed gating strategy refer to Fig. S1 (Supplementary Materials).

Reviewer#3 C20) Line 248 – mechanical lysis was carried out in co-operation with...

Response to reviewer#3 C20: Thank you for the suggestion. We have revised the text accordingly.

Methods, L 266-269: Mechanical lysis of bacterial cultures was carried out in ~~done in kind~~ cooperation with Dr. Robert Hurwitz (MPI Berlin, Germany) via ultrasonification using a Branson Sonifier 450, 400W Titan-Microtip in 4 × 10 cycles (output control: 4, duty cycle: 40%) per culture.

Reviewer#3 C21) Line 350: Re word for clarity and conciseness. - Elevated compared to what?

Response to reviewer#3 C21: We have reworded the text accordingly, please refer also to question/remark 10) (R#3).

Results, L 370-379: As expected, *Pseudomonas*-induced VAP led to an increase in local IL-6 and other pro-inflammatory cytokine levels, including IL-23, TNF α and IL-1 α ~~were elevated~~ in BALF at 24 hpi (Fig. 2a; Supplementary Fig. S2). Following phage therapy, either alone or combined with meropenem, cytokine levels were significantly lower compared to the untreated control group. In contrast, monotherapy with meropenem resulted in a trend toward reduced levels of IL-6 and IL-23 and significantly lowered levels of TNF α and IL-1 α compared to the control group. However, these levels remained higher compared to those observed in the phage-only and combination treatment groups.

Reviewer#3 C22) Line 363 – “As we observed good treatment results” – reword for clarity regarding what results were obtained. Improved clinical scores, rather than reduced bacterial burden.

Response to reviewer#3 C22: Thank you for the remark. We have reworded the text for clarity.

Results, L 392-395: As we observed ~~positive~~ good treatment results at the ~~primary site of infection~~ in terms of clinical improvement with reduced levels of pulmonary inflammatory markers upon phage monotherapy and adjunctive phage treatment, we next investigated lung barrier damage and systemic organ injury.

Reviewer#3 C23) Figure 4c: ‘c’ covered by graph

Response to reviewer#3 C23: We have rearranged the Fig. 4 accordingly.

Reviewer#3 C24) Line 443-444 – phage therapy is more effective. PAO1 were cultured in the absence.

Response to reviewer#3 C24: Thank you for the remark. We have revised the figure legend of Fig. 5 accordingly.

Figure legend, Fig. 5, L 479-482: **Phage-antibiotic combination therapy is more effective in vitro than antibacterial agents alone.** [...] For the infection assay PAO1 were cultured in ~~the~~ absence (control) or presence of meropenem (MIC = 1 μ g/mL), the *Pseudomonas*-specific phage cocktail, or the combination of both in different concentrations as indicated.

Reviewer#3 C25) General suggestions regarding discussion.

- Some commentary on target patient population and why this would be of benefit to these patients.
- Some discussion/reference to overuse of other antibiotic classes- Is there potential scope for phages to have similar synergistic effects with other classes.
- Could this be used in other forms of acute infection Community acquired pneumonia?
- Some commentary required regarding the benefits of combination therapy.

Response to reviewer#3 C25: Thank you for pointing this out.

Regarding your first comment: Phage therapy, especially adjunctive phage treatment, holds great potential for patients under mechanical ventilation dealing with ventilator-associated pneumonia (VAP) and those with other difficult-to-treat infections, such as *Pseudomonas* infections in cystic fibrosis (CF) patients. A recent multicenter, multinational, retrospective observational study demonstrated that personalized phage therapy can lead to significant clinical improvements²⁸. These include eradication of the targeted bacteria, antibiotic re-sensitization, and reduced virulence in bacterial isolates that developed phage resistance during treatment. In addition, our study clearly indicates that phage treatment may help mitigate excessive inflammation and lung barrier destabilization. From a clinical perspective, these findings suggest a potential reduction in complications such as acute lung injury or the more severe acute respiratory distress syndrome (ARDS), which are primarily driven by an exaggerated immune response. This could offer significant benefits for VAP patients at risk of developing this condition.

Regarding your second comment: Indeed, the number of multidrug-resistant (MDR) bacteria continues to rise^{29,30} due to the emergence of resistance against various antibiotic classes, as mentioned in the introduction (L 56 ff). Therefore, developing new therapeutic approaches remains crucial. However, there is a high incidence of synergistic effects when phages are combined with various antibiotics, highlighting the (clinical) potential of adjuvant phage therapy²⁸.

Regarding your third comment: We fully agree that it is an interesting and important aspect to discuss. Phage therapy, particularly adjunctive phage therapy, has potential applications for other bacterial infections. For instance, it has already been demonstrated as a promising novel treatment strategy for multidrug-resistant (MDR) *Klebsiella pneumoniae*, one of the most common pathogens responsible for community-acquired and nosocomial infections^{31,32}.

Regarding your fourth comment: We found that combination therapy offers significant benefits over monotherapies, improving therapeutic outcomes. Adjunctive phage therapy enhances treatment effectiveness through the synergistic interaction between bacteriophages and antibiotics. The personalized selection of phage cocktails combined with conventional antibiotics further optimizes efficacy by targeting specific bacterial strains.

To address all comments more detailed in the discussion section of our manuscript, we have revised the text accordingly.

Discussion, L 647-653: A recently published multicentre, multinational, retrospective observational study evidenced numerous examples of synergistic or additive interactions between phages and antibiotics, both *in vivo* and *in vitro*, leading to significant clinical improvements⁵⁸. Treatment with an intravenous phage cocktail combined with systemic antibiotics from different classes successfully eradicated multidrug-resistant (MDR) *P. aeruginosa* pneumonia in a cystic fibrosis (CF) patient, restoring health and enabling lung transplantation²⁷.

Discussion, L 783-794: In summary, adjunctive phage therapy holds significant potential for mechanically ventilated patients with ventilator-associated pneumonia (VAP). While our murine VAP study focused on *P. aeruginosa*, (adjunctive) phage therapy is also suitable for other MDR bacterial infections, such as *Klebsiella pneumoniae*, a major cause of both community-acquired and nosocomial infections^{85,86}. In general, combination therapy offers key advantages over monotherapies, including the mitigation of excessive inflammation and associated lung barrier destabilization, as demonstrated in our study. Additionally, improved eradication of MDR bacteria, restoration of antibiotic sensitivity, and reduced bacterial virulence in phage-resistant strains offer further advantages. These effects could translate into significant clinical benefits, potentially limiting acute lung injury while reducing the side effects of conventional antibacterial therapies and minimizing antibiotic overuse, thereby supporting a safer and more effective treatment option for future clinical applications.

Reviewer #4 (Remarks to the Author):

Response to reviewer#4: We would like to sincerely thank you for reviewing our manuscript.

References

1. Debarbieux, L. *et al.* Bacteriophages can treat and prevent *Pseudomonas aeruginosa* lung infections. *The Journal of infectious diseases* **201**, 1096–1104; 10.1086/651135 (2010).
2. Felten, M. *et al.* Overventilation-induced airspace acidification increases susceptibility to *Pseudomonas pneumonia* **22**; 10.1101/2024.08.07.603041 (2024).
3. Verbrugge, S. J. *et al.* Lung overinflation without positive end-expiratory pressure promotes bacteremia after experimental *Klebsiella pneumoniae* inoculation. *Intensive care medicine* **24**, 172–177; 10.1007/s001340050541 (1998).
4. Dhanireddy, S. *et al.* Mechanical ventilation induces inflammation, lung injury, and extra-pulmonary organ dysfunction in experimental pneumonia. *Laboratory investigation; a journal of technical methods and pathology* **86**, 790–799; 10.1038/labinvest.3700440 (2006).
5. Tsay, T.-B., Jiang, Y.-Z., Hsu, C.-M. & Chen, L.-W. *Pseudomonas aeruginosa* colonization enhances ventilator-associated pneumonia-induced lung injury. *Respiratory research* **17**, 101; 10.1186/s12931-016-0417-5 (2016).
6. Robak, O. H. *et al.* Antibiotic treatment-induced secondary IgA deficiency enhances susceptibility to *Pseudomonas aeruginosa pneumonia*. *The Journal of clinical investigation* **128**, 3535–3545; 10.1172/JCI97065 (2018).
7. Weissfuss, C. *et al.* Repetitive Exposure to Bacteriophage Cocktails against *Pseudomonas aeruginosa* or *Escherichia coli* Provokes Marginal Humoral Immunity in Naïve Mice. *Viruses* **15**; 10.3390/v15020387 (2023).
8. Cisneros, J. M. *et al.* Colistin versus meropenem in the empirical treatment of ventilator-associated pneumonia (Magic Bullet study): an investigator-driven, open-label, randomized, noninferiority controlled trial. *Critical care (London, England)* **23**, 383; 10.1186/s13054-019-2627-y (2019).
9. Mangioni, D. *et al.* Incidence, microbiological and immunological characteristics of ventilator-associated pneumonia assessed by bronchoalveolar lavage and endotracheal aspirate in a prospective cohort of COVID-19 patients: CoV-AP study. *Critical care (London, England)* **27**, 369; 10.1186/s13054-023-04658-5 (2023).
10. Oshima, K. *et al.* Efficacy of High-Dose Meropenem (Six Grams per Day) in Treatment of Experimental Murine Pneumonia Induced by Meropenem-Resistant *Pseudomonas aeruginosa*. *Antimicrobial agents and chemotherapy* **61**; 10.1128/AAC.02056-16 (2017).
11. Yamada, K. *et al.* In vivo efficacy and pharmacokinetics of biapenem in a murine model of ventilator-associated pneumonia with *Pseudomonas aeruginosa*. *Journal of infection and chemotherapy : official journal of the Japan Society of Chemotherapy* **18**, 472–478; 10.1007/s10156-011-0359-2 (2012).
12. Benavent, E. *et al.* Efficacy of meropenem extended infusion vs intermittent bolus monotherapy and in combination with colistin against *Pseudomonas aeruginosa* biofilm. *International journal of antimicrobial agents* **62**, 106856; 10.1016/j.ijantimicag.2023.106856 (2023).
13. Landersdorfer, C. B. *et al.* Optimization of a Meropenem-Tobramycin Combination Dosage Regimen against Hypermutable and Nonhypermutable *Pseudomonas aeruginosa* via Mechanism-Based Modeling and the Hollow-Fiber Infection Model. *Antimicrobial agents and chemotherapy* **62**; 10.1128/AAC.02055-17 (2018).

14. Grace, A., Sahu, R., Owen, D. R. & Dennis, V. A. *Pseudomonas aeruginosa* reference strains PAO1 and PA14: A genomic, phenotypic, and therapeutic review. *Frontiers in microbiology* **13**, 1023523; 10.3389/fmicb.2022.1023523 (2022).
15. Kuang, Z. *et al.* Surfactant phospholipids act as molecular switches for premature induction of quorum sensing-dependent virulence in *Pseudomonas aeruginosa*. *Virulence* **11**, 1090–1107; 10.1080/21505594.2020.1809327 (2020).
16. Cabrol, S., Olliver, A., Pier, G. B., Andremont, A. & Ruimy, R. Transcription of quorum-sensing system genes in clinical and environmental isolates of *Pseudomonas aeruginosa*. *Journal of bacteriology* **185**, 7222–7230; 10.1128/JB.185.24.7222-7230.2003 (2003).
17. Ferran, A. A. *et al.* The Selection of Antibiotic- and Bacteriophage-Resistant *Pseudomonas aeruginosa* Is Prevented by Their Combination. *Microbiology spectrum* **10**, e0287422; 10.1128/spectrum.02874-22 (2022).
18. Valério, N. *et al.* Effects of single and combined use of bacteriophages and antibiotics to inactivate *Escherichia coli*. *Virus research* **240**, 8–17; 10.1016/j.virusres.2017.07.015 (2017).
19. Bürkle, M. *et al.* Phage-phage competition and biofilms reduce the efficacy of a combination of two virulent bacteriophages against *Pseudomonas aeruginosa*; 10.1101/2024.09.13.612609 (2024).
20. Garbe, J. *et al.* Characterization of JG024, a *Pseudomonas aeruginosa* PB1-like broad host range phage under simulated infection conditions. *BMC microbiology* **10**, 301; 10.1186/1471-2180-10-301 (2010).
21. Selezska, K. *et al.* *Pseudomonas aeruginosa* population structure revisited under environmental focus: impact of water quality and phage pressure. *Environmental microbiology* **14**, 1952–1967; 10.1111/j.1462-2920.2012.02719.x (2012).
22. Dequin, P.-F. *et al.* Hydrocortisone in Severe Community-Acquired Pneumonia. *The New England journal of medicine* **388**, 1931–1941; 10.1056/NEJMoa2215145 (2023).
23. Heidari, S., Babor, T. F., Castro, P. de, Tort, S. & Curno, M. Sex and Gender Equity in Research: rationale for the SAGER guidelines and recommended use. *Research integrity and peer review* **1**, 2; 10.1186/s41073-016-0007-6 (2016).
24. Morgan, R. & Klein, S. L. The intersection of sex and gender in the treatment of influenza. *Current opinion in virology* **35**, 35–41; 10.1016/j.coviro.2019.02.009 (2019).
25. Dias, S. P., Brouwer, M. C. & van de Beek, D. Sex and Gender Differences in Bacterial Infections. *Infection and immunity* **90**, e0028322; 10.1128/iai.00283-22 (2022).
26. Sophia Charis Rotter. Studying *Pseudomonas aeruginosa* infection and therapeutic approaches in in vitro lung models. Master thesis. Technical University Berlin, 2024.
27. Fischer, S., Kittler, S., Klein, G. & Glünder, G. Microplate-test for the rapid determination of bacteriophage-susceptibility of *Campylobacter* isolates-development and validation. *PLoS one* **8**, e53899; 10.1371/journal.pone.0053899 (2013).
28. Pirnay, J.-P. *et al.* Personalized bacteriophage therapy outcomes for 100 consecutive cases: a multicentre, multinational, retrospective observational study. *Nature microbiology* **9**, 1434–1453; 10.1038/s41564-024-01705-x (2024).
29. Rosenthal, V. D. *et al.* International Nosocomial Infection Control Consortium (INICC) report, data summary of 45 countries for 2013-2018, Adult and Pediatric Units, Device-

- associated Module. *American journal of infection control* **49**, 1267–1274; 10.1016/j.ajic.2021.04.077 (2021).
30. World Health Organization. WHO bacterial priority pathogens list, 2024: Bacterial pathogens of public health importance to guide research, development and strategies to prevent and control antimicrobial resistance. Available at <https://www.who.int/publications/i/item/9789240093461> (2024).
 31. Zaki, B. M., Hussein, A. H., Hakim, T. A., Fayez, M. S. & El-Shibiny, A. Phages for treatment of *Klebsiella pneumoniae* infections. *Progress in molecular biology and translational science* **200**, 207–239; 10.1016/bs.pmbts.2023.03.007 (2023).
 32. Wang, Z. *et al.* Combination Therapy of Phage vB_KpnM_P-KP2 and Gentamicin Combats Acute Pneumonia Caused by K47 Serotype *Klebsiella pneumoniae*. *Frontiers in microbiology* **12**, 674068; 10.3389/fmicb.2021.674068 (2021).

REVIEWERS' COMMENTS #2 – POINT BY POINT REPLY

Response: We sincerely thank the editor and, in particular, the reviewers for their time and thoughtful feedback. We are grateful that the reviewers recognize our efforts to improve the manuscript based on their guidance and agree that, with their support, it has significantly improved.

Reviewer #1 (Remarks to the Author):

I appreciate the authors' efforts in revising the manuscript. The point-by-point responses are clear and adequately address the concerns I previously raised. At this stage, I have only minor suggestions for further improvement.

Figure 1: Did the authors compute the clinical score at extubation (0 hpi)? If so, these results should be included in panels (b), (c), and (d).

Response: Many thanks for this suggestion. This would indeed be an insightful time point to have a clinical score. However, we did not record these values as they would be confounded by the lingering effects of just antagonized anesthesia.

Figure 5 / Supplementary Figure S5: Please include PFU/ml data to assess the effect of antibiotics on phage replication.

Response: Indeed, this would have been an informative read-out in this experimental setting and was one we considered during the design phase. However, due to the limited amount of sample material generated under this experimental layout, our focus remained on CFU/mL and endotoxin measurements. In particular, the endotoxin analysis required the majority of the sample, leaving insufficient material for PFU measurements. The rapid bactericidal effect of the phages was evident in the reduced CFU/mL values, as well as in the observed re-growth of likely phage-resistant bacterial clones under phage monotherapy. Regarding endotoxin release by the different antibacterial treatments, assessing PFU/mL in the phage-treated conditions would not have provided additional insight.

These minor additions will enhance the clarity and completeness of the manuscript.

Thank you for your thoughtful revisions. I look forward to the final version of the manuscript.

Reviewer #3 (Remarks to the Author):

I am satisfied that all concerns raised during peer review have been addressed in the revisions, which has greatly improved the manuscript.

Response: Many thanks. We are happy to hear that.